# Structural insights into strigolactone catabolism by carboxylesterases reveal a conserved conformational regulation

Malathy Palayam [1], Linyi Yan[1], Ugrappa Nagalakshmi[1], Amelia K. Gilio[1], David Cornu[2], François-Didier Boyer [3], Savithramma P. Dinesh-Kumar [1,4] & Nitzan Shabek [1] ✉

Phytohormone levels are regulated through specialized enzymes, participating not only in their biosynthesis but also in post-signaling processes for signal inactivation and cue depletion. *Arabidopsis thaliana* (At) carboxylesterase 15 (CXE15) and carboxylesterase 20 (CXE20) have been shown to deplete strigolactones (SLs) that coordinate various growth and developmental processes and function as signaling molecules in the rhizosphere. Here, we elucidate the X-ray crystal structures of AtCXE15 (both apo and SL intermediate bound) and AtCXE20, revealing insights into the mechanisms of SL binding and catabolism. The N-terminal regions of CXE15 and CXE20 exhibit distinct secondary structures, with CXE15 characterized by an alpha helix and CXE20 by an alpha/ beta fold. These structural differences play pivotal roles in regulating variable SL hydrolysis rates. Our findings, both in vitro and in planta, indicate that a transition of the N-terminal helix domain of CXE15 between open and closed forms facilitates robust SL hydrolysis. The results not only illuminate the distinctive process of phytohormone breakdown but also uncover a molecular architecture and mode of plasticity within a specific class of carboxylesterases.

Strigolactones (SL) serve as pivotal plant hormones, orchestrating numerous developmental processes, such as the control of shoot branching and root architecture, as well as functioning as secreted signals in the rhizosphere, enabling communication with both symbiotic fungi and parasitic plants[1–4]. SLs are derived from β-carotene and are classified into distinct canonical and non-canonical groups. Canonical SLs feature a characteristic tricyclic lactone structure known as the ABC-ring connected to a bute-nolide D-ring through an enol ether bridge, while non-canonical SLs do not exhibit the conventional ABC-ring structures yet retain the same bridge with the D-ring (Supplementary Fig. 1a)[5–8]. SLs are perceived by the multifunctional receptor/hydrolase DWARF14 (D14), which subsequently recruits the ubiquitin ligase DWARF3 (D3 in rice) or MORE AXILLARY GROWTH2 (MAX2) F-box

protein to target transcriptional repressor DWARF53 (in rice)/ SUPPRESSOR OF MAX2-1 Like proteins (SMXL6, SMXL7, and SMXL8), leading to proteasome-mediated degradation and the subsequent expression of SL responsive signaling genes[9–12].

Upon SL binding, the receptor D14 activates the signaling pathway by hydrolyzing the SL[11,13], releasing the ABC-ring, and covalently linking the modified D-ring to the D14 catalytic site[13–15]. Thus, D14 deactivates SL through irreversible catalysis and is thought to consequently cause local depletions of bioactive SLs[16,17]. Nonetheless, D14s are broadly characterized as futile hydrolytic enzymes with relatively slow turnover rates ($k_{cat} \sim 0.33$ min$^{-1}$ and $K_m = 0.43$ μM) designed to facilitate conformational changes for the recruitment of key SL signaling components, rather than functioning to regulate SL homeostasis levels[13,15,17–19].

[1]Department of Plant Biology, College of Biological Sciences, University of California—Davis, Davis, CA, USA. [2]Institute for Integrative Biology of the Cell (I2BC), Université Paris-Saclay, CEA, CNRS, Gif-sur-Yvette, France. [3]Institut de Chimie des Substances Naturelles, Université Paris-Saclay, CNRS UPR 2301, Gif-sur-Yvette, France. [4]The Genome Center, University of California—Davis, Davis, CA, USA. ✉e-mail: nshabek@ucdavis.edu

Ensuring the precise regulation of various phytohormone levels is imperative, encompassing not only their production through biosynthesis pathways and transporters to elevate local concentrations but also the deactivation and depletion of bioactive molecules post-signaling through specialized enzymes. For instance, dioxygenase auxin oxidation regulates the levels of Auxin homeostasis[20]; GA2-oxidases enzymes deactivate gibberellic acid (GA)[21,22]; CYP707A enzymes modify abscisic acid (ABA)[23]; members of cytochromes P450 family CYP94 and CYP72B1 deactivate Jasmonyl-L-isoleucine[24–26] and brassinosteroids (BR)[27]; and downy mildew resistance 6 is implicated in hydroxylating salicylic acid[28]. Additionally, enzymes such as SABATH methyltransferases and GH3 (GRETCHEN HAGEN 3) acyl amido synthetases are known to modulate the amounts of various phytohormones by making chemical modification, thereby contributing to their homeostasis[29–31]. Despite the growing number of studies that have uncovered enzymes involved in phytohormone post-signaling deactivation, numerous questions remain unanswered regarding the identification, biochemical characterization, and regulation of such enzymes in signaling pathways.

Notably, recent studies in *Arabidopsis* revealed the involvement of carboxylesterase 15 (CXE15) and carboxylesterase 20 (CXE20) enzymes in SL homeostasis[32,33]. The carboxylesterase (CXE) family represents a subset of the α/β hydrolase superfamily of proteins, which typically feature a central core, comprising 8–11 β-strands, enveloped by α-helices and catalyze the hydrolysis of carboxyl esters into alcohol and carboxylate products[34,35]. The role of CXE15 in SL signaling regulation was investigated through overexpression (*CXE15-OE*) and knockout mutants. *CXE15-OE* plants resembled *d14* SL receptor mutants and showed similar transcriptome profiles, particularly suppressing *BRANCHED1* (*BRC1*) expression, while *cxe15* knockout plants were normal[33]. Additionally, the transcript levels of *AtCXE15* were upregulated when axillary buds were treated with exogenously applied bioactive canonical and non-canonical SLs. In a hypocotyl elongation assay, *AtCXE15-OE* and *d14* seedlings showed reduced sensitivity to SL treatment, whereas *cxe15* mutants were more sensitive. Moreover, grafting SL biosynthetic mutants (*atmax3-9*, *atmax4-1*, and *atmax1-1*) onto *AtCXE15-OE* and *atcxe15-1* rootstocks showed that *atcxe15-1* rootstocks restored the hyperbranching phenotype to wild type. Grafting *atmax3-9* shoots onto *AtCXE15-OE* rootstocks inhibited excessive branching, while *atmax4-1* and *atmax1-1* shoot branching was not fully rescued. This indicates that overaccumulation of AtCXE15 in *Arabidopsis* may cause a deficiency in SL, as well as non-canonical SLs like CL, carlactone (CLA), and/or methyl carlactonoate (MeCLA) derivatives[33].

Both CXE15 and D14 catalyze the same reaction, ultimately cleaving the enol ether bridge between the butenolide moiety (D-ring) and the ABC-ring. Nonetheless, CXE15 displays robust hydrolytic activity on various SLs, including CLA, at a significantly higher rate than D14[33], therefore confirming that CXE15 may serve as a bona fide SL catabolizing enzyme. Another carboxylesterase, CXE20 was identified during a drought-tolerant screening while ectopically expressed in maize, revealing phenotypes that typically are associated with SL-deficient mutants[32]. Similarly, it was suggested that CXE20 can deplete SL by sequestering the free phytohormone, yet it is unclear whether SL can be catabolized by the CXE20 class of carboxylesterases. Despite both CXE15 and CXE20 being classified under the carboxylesterase superfamily, they exhibit low sequence similarity, making it challenging to predict their precise molecular functions. Moreover, the overall biochemical characterization of carboxylesterases has been underexplored in planta compared to other kingdoms of life. Therefore, the characterization, mode of action, and molecular structures of CXEs in regulating SL homeostasis remain to be elucidated.

In this study, we analyzed twenty *Arabidopsis thaliana* (*At*) CXE proteins through phylogenetic analysis, revealing the presence of a distinct N-terminal (NT) region across various CXE clades. We further determined the X-ray crystal structures of AtCXE15$_{apo}$, AtCXE15-SL (CXE15$^{GR24}$), and AtCXE20 at 1.8–2.3 Å resolution, and provided structural insights into SL binding and catabolism. Our structural analysis confirms the presence of distinct yet evolutionarily conserved NT domains between CXE15 (NT helix, NTH) and CXE20 (NT α/β fold). Importantly, through in-planta, biochemical, and in silico analyses, we demonstrate that, unlike CXE20, the NTH domain of CXE15 undergoes a dynamic transition from an open to a closed state, thereby facilitating robust SL hydrolysis. These findings underscore the functional variations in the regulation of SL homeostasis in plant tissues. Together, this study provides structural insights into a unique phytohormone breakdown process in the plant kingdom and unveils a molecular architecture and mode of plasticity of a specific class of carboxylesterases.

## Results

### Sequence analysis of carboxylesterases reveals three distinct clades and diverged NT region

To elucidate the structure-function of CXEs in the SL homeostasis regulation, we first examined the evolutionary relationships among the previously annotated 20 CXEs from *Arabidopsis*. CXEs can be grouped into three distinct clades based on the phylogenetic tree generated using full-length protein sequences (Fig. 1a). Notably, clade I and clade III are subdivided into two subgroups, with 17 of the encoded gene family members residing in these clades, while clade II consists of three members. Further analysis highlighted the presence of conserved consensus motifs at the catalytic core region of the enzymes, such as His-Gly-Gly-Gly (HGGG, corresponding to positions 83–86 in AtCXE15) and Gly-X-Ser-X-Gly (GXSXG at position 167–171 in CXE15) (Supplementary Fig. 1b). These motifs are known to participate in the oxyanion hole formation and are crucial for stabilizing the tetrahedral intermediate transition state during catalysis[36–39]. Within the GXSXG motif, the central serine is predicted to be part of the catalytic triad, and the residue identities at the 'X' positions have served to further classify the esterase into either the Gly-Asp-Ser-Ala-Gly (GDSAG) or Gly-Thr-Ser-Ala-Gly (GTSAG) subgroup[40,41]. Based on this, clades I (CXE1/2/5/7/12/13) and II (CXE6/15/17) belong to the GDSAG subgroup. Whereas clade III encompasses GDSSG (CXE10/14/19), GVSCG (CXE11/16), GSSSG (CXE8), GSSNG (CXE9), and GTSAG (CXE20) subgroups (Supplementary Figs. 1b and 2a). Notably, the GDSSG motif is indicative of non-functional carboxylesterases, represented by GA receptors, GID1a/b/c, in which the catalytic histidine is replaced by valine or isoleucine. Comparing the Ser-His-Asp/Glu catalytic triad sequence of all 20 CXEs revealed that in CXE15 aspartic acid is replaced by glutamic acid (Supplementary Fig. 2a), making it the only CXE enzyme that is similar to the carboxylesterases in the animal kingdom[42]. Strikingly, the comparison of all 20 CXEs revealed major variations in the NT region (1-50 amino acids) across all clades (Supplementary Fig. 1b). AtCXE4 in clade I is the only enzyme featuring an NT mitochondrial transit peptide, suggesting that the diversity observed in CXE clades is not directly linked to subcellular localization function. Building upon a previous study proposing the role of CXEs in SL depletion[32,33], we analyzed CXE15 and CXE20 across land plant species, that represent clades II and III. Phylogenetic analysis of CXE15 subdivided the lineage into monocots versus dicots (Fig. 1b) and further revealed conserved sequence divergence in the NT region between these subgroups (Supplementary Fig. 2b). Sequence analysis of CXE20 also showed a highly conserved NT region in all dicots (Supplementary Fig. 2c-d). Together, our analysis indicates a diverged NT region in CXE15 and CXE20 may contribute to their distinct functions in plants.

### Crystal structures of CXE15 and CXE20 reveal distinct NT domains

To further investigate whether the NT region has any structural and functional significance, we purified and determined the crystal

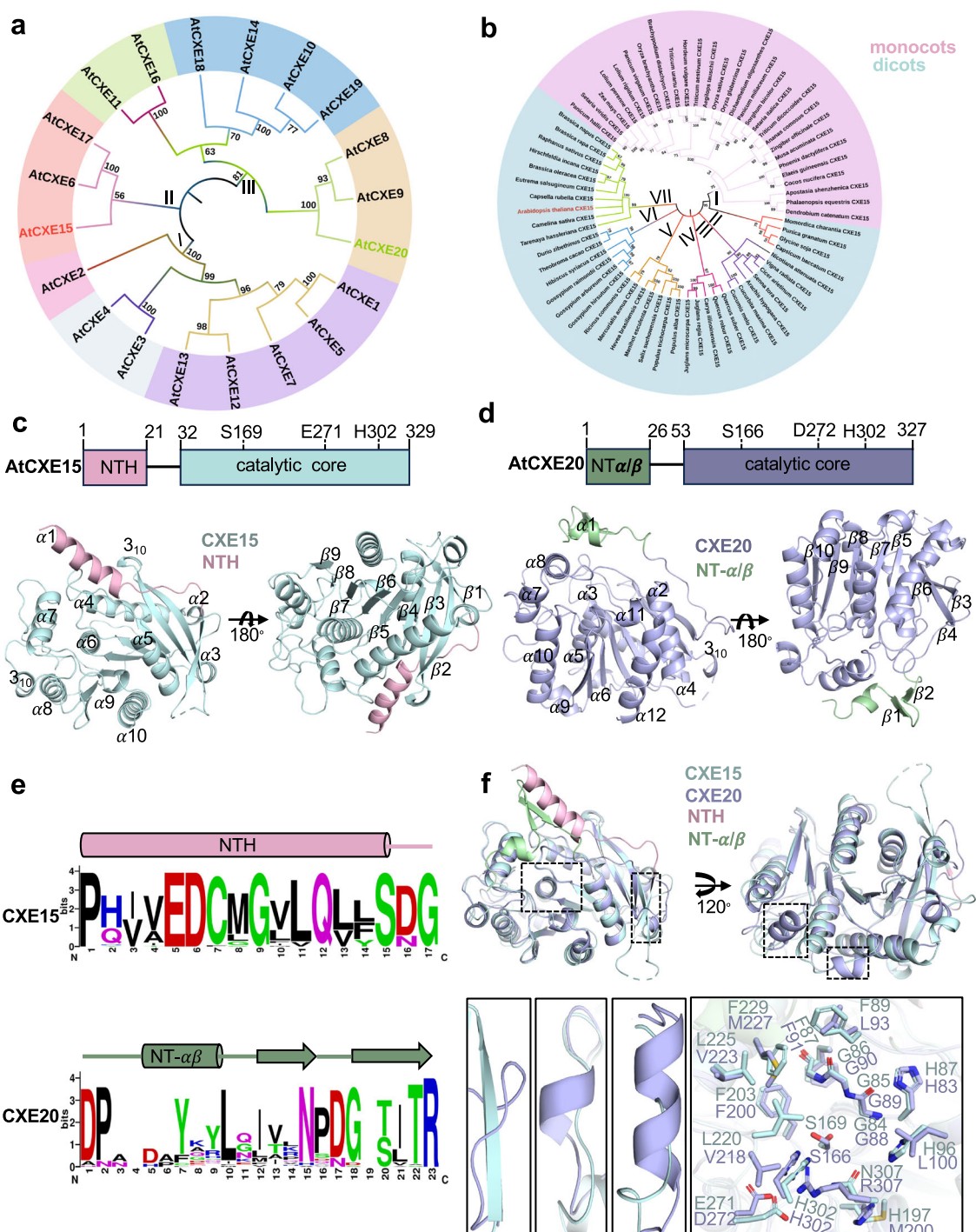

**Fig. 1 | CXE sequence analysis and crystal structures of AtCXE15 and AtCXE20.**
**a, b** Phylogenetic tree of At CXEs based on protein sequence alignment. CXEs within the same clade are highlighted with distinctive background colors (CXE15 and CXE20 in red and green) **a**. Phylogenetic tree of CXE15 based on protein sequences retrieved from NCBI. The diversification of CXE15 between the monocots and dicots is shown in pink and cyan shades (**b**). All sequence alignments were constructed by MEGA11 software using a maximum likelihood algorithm. Values on the nodes represent the bootstrap replicates of 1000 and values greater than 50% are shown. **c, d** Schematic representation of AtCXE15 and AtCXE20 core domains. Crystal structure of AtCXE15 (pale cyan) showing α/β hydrolase fold with distinct NTH (light pink) shown in different angles (**c**). Crystal structure of AtCXE20 (light blue) showing α/β hydrolase fold with distinct NT α/β helix (NT-α/β, light green) (**d**). **e** The sequence logo of the NT region of AtCXE15 (top) and AtCXE20 (bottom) represents the amino acid conservation between CXE15 and CXE20 proteins across land plants. **f** Comparative structural analysis and superposition of AtCXE15 (pale cyan) and AtCXE20 (light blue) shown in different angles (top). Close-up view of the catalytic cavity (bottom-right) and the distinct regions (bottom-left) between CXE15 and CXE20 (bottom). Source data are provided as a Source Data file.

structures of AtCXE15 and AtCXE20 at 2.3 Å and 1.8 Å resolution, respectively (Fig. 1c, d, Supplementary Fig. 3, and Table 1). The overall architecture of AtCXE15 is divided into two domains: a cap domain (residues 1–21) and a core catalytic domain (residues 32–329), both of which are connected by a long linker loop (residues 22–31) (Fig. 1c). The cap domain of CXE15 includes the NT α-helix (α1, we termed NTH) which is highly conserved among all CXE15 variants (Fig. 1e and Supplementary Fig. 2b), while the catalytic domain adopts a common α/β

## Table 1 | Data collection and refinement statistics

| | AtCXE15 | AtCXE15$^{GR24}$ | AtCXE20 |
|---|---|---|---|
| **Data collection** | | | |
| Space group | P 4$_1$2$_1$2 | P 4$_1$2$_1$2 | P 2$_1$2$_1$2$_1$ |
| Cell dimensions | | | |
| *a, b, c* (Å) | 83.9, 83.9, 117.1 | 84.3, 84.3, 117.4 | 66.6, 95.7, 100.3 |
| Resolution (Å) | 48.0–2.3 (2.38–2.30)$^a$ | 42.1–2.3 (2.40–2.32)$^a$ | 48–1.8 (1.91–1.85)$^a$ |
| $R_{sym}$ | 0.049 (0.257) | 0.034 (0.288) | 0.035 (0.230) |
| $I/\sigma I$ | 11.0 (2.16) | 18.5 (2.0) | 23.2 (2.5) |
| Completeness (%) | 99.95 (99.84) | 100.0 (100.0) | 99.82 (99.08) |
| Redundancy | 9.2 (8.5) | 24.7 (17.4) | 12.6 (12.5) |
| **Refinement** | | | |
| Resolution (Å) | 2.30 | 2.32 | 1.85 |
| No. reflections | 19200 | 18536 | 55300 |
| $R_{work}/R_{free}$ (%) | 20.8/24.4 | 18.9/22.9 | 17.1/19.9 |
| No. atoms | 2581 | 2611 | 5356 |
| Protein | 2515 | 2535 | 4948 |
| Ligand/ion | 18 | 26 | 18 |
| Water | 48 | 54 | 390 |
| *B*-factors | 35.8 | 43.4 | 25.7 |
| Protein | 35.8 | 43.3 | 25.2 |
| Ligand/ion | 43.1 | 55.6 | 37.1 |
| Water | 33.8 | 41.9 | 31.2 |
| R.m.s. deviations | | | |
| Bond lengths (Å) | 0.003 | 0.003 | 0.008 |
| Bond angles (°) | 0.65 | 0.54 | 1.00 |
| Ramachandran plot analysis | | | |
| Favored region (%) | 96.52 | 96.85 | 97.08 |
| Additionally allowed (%) | 3.48 | 3.15 | 2.92 |
| Disallowed region (%) | 0.00 | 0.00 | 0.00 |
| PDB ID | 8VCA | 8VCD | 8VCE |

$^a$Values in parentheses are for the highest-resolution shell.

hydrolase fold with a central nine-stranded β-sheet surrounded by ten α-helices and shares significant homology with rest of the members of CXEs (Fig. 1c and Supplementary Fig. 4). The catalytic triad consists of residues S169, H302, and E271, where the catalytic serine is positioned at the apex between α6 and β6. E271 is situated in the loop between α8 and β8 while H302 is in the loop between β9 and α10 (Fig. 1c and Supplementary Fig. 4). The oxyanion hole is located above the active site within the conserved HGGGF motif, positioned in the loop between α5 and β5 (Supplementary Fig. 4). Notably, this motif plays a crucial role in stabilizing the high-energy oxyanion intermediate through a hydrogen bond interaction with the substrate carbonyl group.

Similar to AtCXE15, the overall structure of AtCXE20 is divided into two domains: cap domain (residues 1–26) and the catalytic core domain (residues 53–327) both of which are also connected by a long linker loop (residues 27–52) (Fig. 1d). The catalytic core of AtCXE20 adopts the common α/β hydrolase fold with eight-stranded β-sheet surrounded by 11 α-helices and one 3$_{10}$ helix (Fig. 1d). Unlike CXE15, CXE20 features a catalytic triad composed of Ser-His-Asp (instead of Ser-His-Glu), and both enzymes exhibit similar catalytic pockets, predominantly lined with hydrophobic residues (Fig. 1f). Strikingly, in contrast to AtCXE15, the cap domain of AtCXE20 comprises of NT α1 helix and anti-parallel β-sheet (β1 and β2), which we termed NT-α/β

(Fig. 1d, f and Supplementary Fig. 5). Moreover, the loop connecting the NT-α/β to the core is longer in CXE20 compared to the loop connecting the NTH in AtCXE15 that is replaced by an extra β-strand (Fig. 1f). Additionally, CXE20 displays a short helix that connects the α9 and β9 regions, whereas this region is unstructured in CXE15.

In addition to these significant variations, our comparative analysis between AtCXE15, AtCXE20, and other plant carboxylesterases, including AeCXE1 and OsGID1 a/b/c (classified under CXE10, CXE14, and CXE19), has elucidated a conserved core region characterized by the α/β hydrolase fold. Notable differences were observed mostly in their NT domains (Supplementary Fig. 6a, b), underscoring, yet again, the pivotal role of these regions. A broader comparative structural analysis using representative crystal structures of plant and animal CXE/CES showed that plant CXEs differ from animal CES, retaining the catalytic domain and Ser-His-Glu catalytic triad within a deep hydrophobic pocket (Supplementary Fig. 6c). Human and mouse carboxylesterase (CES1) comprises two additional domains: a regulatory domain and an α/β domain, where the regulatory domain serves as surface-binding sites for various substrates, thereby regulating substrate access to the catalytic gorge[43–45]. Altogether, this comparative analysis highlights the substantial regulatory role of the NT region in these enzymes, showcasing CXE15 as a modular enzyme previously unseen in this class of CXE in plants.

### CXE15 and CXE20 can bind SL but only CXE15 functions in effective SL catabolism

To investigate the binding and hydrolysis of SL by CXE15 and CXE20, we first employed differential scanning fluorimetry (DSF), allowing us to evaluate the change in stability of the enzymes upon SL binding. In the presence of SL analog, *rac*-GR24 (Supplementary Fig. 1a), DSF analysis showed a modest thermal shift for both AtCXE15 and AtCXE20 with an increase in melting temperature ($\Delta T_m = +1.0$ °C) observed for AtCXE15 and a decrease in melting temperature ($\Delta T_m = -5.0$ °C) for AtCXE20 (Fig. 2a, b). This indicates that both carboxylesterases can bind SL, leading to a mild stabilization/destabilization of the enzymes as previously demonstrated for SL receptor, D14[17,46–48]. To further substantiate our DSF analysis, we performed trypsin limited proteolytic digestion assay in the presence and absence of GR24. Our results show that the binding of SL analog to CXE15 and CXE20 induces protein stabilization and destabilization, respectively (Fig. 2c). Interestingly, AtCXE20 seems to experience more significant destabilization upon binding GR24, possibly due to conformational changes occurring during perception.

Next, we examined the enzymatic activity of CXE15 and CXE20 using the pro-fluorescent probe Yoshimulactone Green (YLG), which is commonly employed for the measurement of SL hydrolysis[9,18]. CXE15 exhibited significant hydrolytic activity compared to the catalytic mutants CXE15$^{S169A}$ and CXE15$^{E271A}$, which were unable to hydrolyze YLG (Fig. 2d). Notably, CXE20 exhibited minimal activity in comparison to CXE15 (Fig. 2e), suggesting that CXE15 is indeed a highly efficient enzyme for SL depletion, as previously proposed in planta[33]. CXE15 demonstrated much greater activity towards YLG hydrolysis than AtCXE20 with $k_{cat}$ values of 0.84 s$^{-1}$ and 0.021 s$^{-1}$, respectively (Fig. 2f). Interestingly, AtCXE15 exhibited a 2.5-fold decrease in binding affinity to YLG than AtCXE20 with $K_m$ values of $21 \pm 4.7$ μM and $8.6 \pm 2.3$ μM, respectively (Fig. 2f). Despite of the variations in SL binding affinities, AtCXE15 demonstrates superior enzymatic performance in YLG catalysis, with remarkable catalytic efficiency of $4.0 \times 10^{-2}$ s$^{-1}$ μM$^{-1}$, while AtCXE20 at only 6% of this efficiency, measuring just $0.25 \times 10^{-2}$ s$^{-1}$ μM$^{-1}$ (Fig. 2f).

Given the overall architecture similarity between CXE15 and CXE20 catalytic cores, we inspected the SL binding pocket of both enzymes. We found significant changes in the dimensions and biochemical compositions of the pocket where CXE15 exhibits a relatively

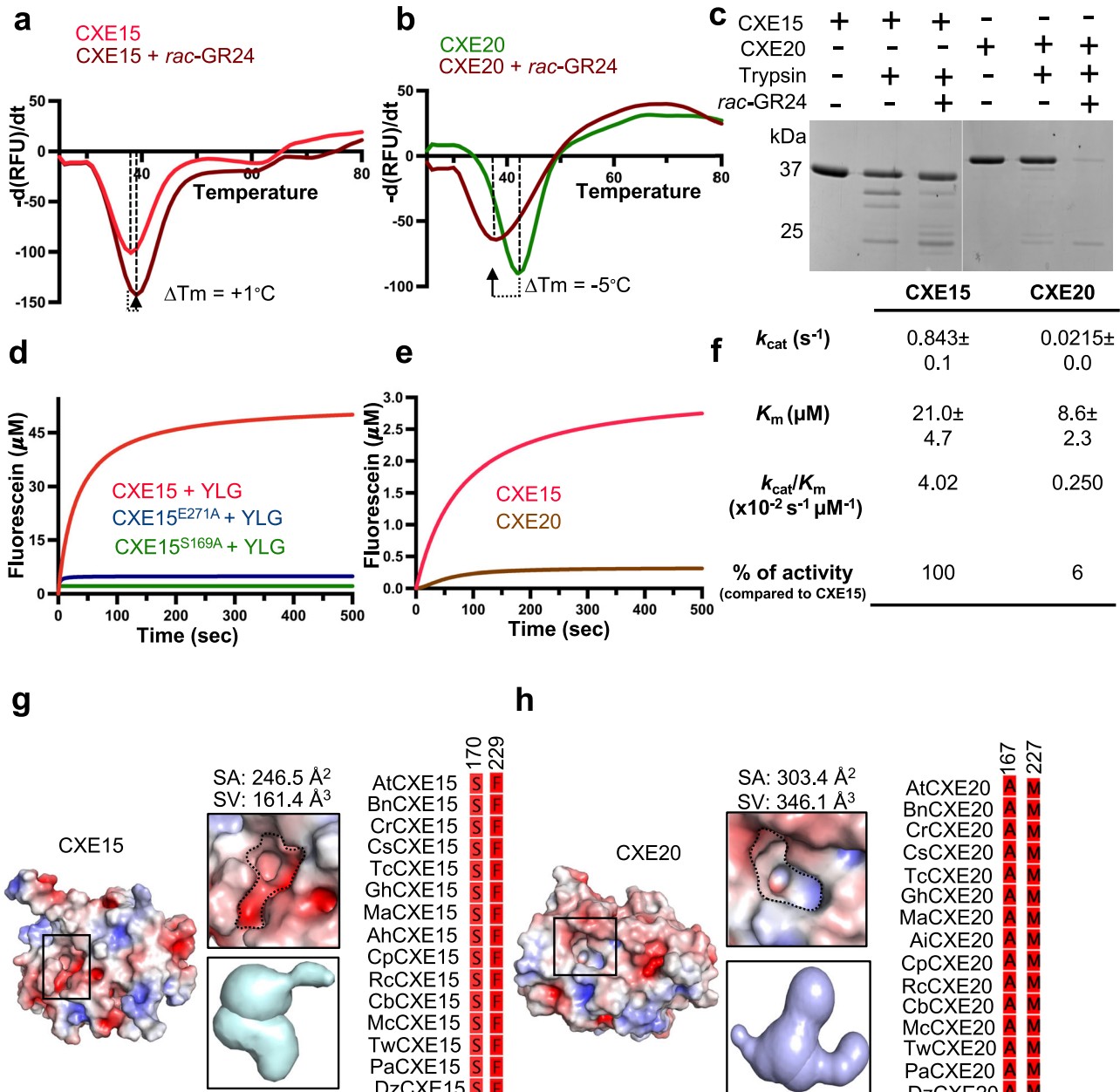

**Fig. 2 | Functional characterization of AtCXE15 and AtCXE20. a, b** Melting temperature curves of AtCXE15 (**a**, red) and AtCXE20 (**b**, green) in the presence or absence of *rac*-GR24 (brown) were determined through DSF analysis. The graph represents the average of three technical replicates. **c** Limited trypsin digestions of AtCXE15 and AtCXE20 are shown in the presence or absence of *rac*-GR24. All proteins were resolved via SDS-PAGE and visualized by Coomassie stain. **d** YLG hydrolysis assay of AtCXE15 (red), AtCXE15$^{E271A}$ (blue), and AtCXE15$^{S169A}$ (green). **e** Comparative YLG hydrolysis assay of CXE15 (red) and CXE20 (brown). Colored lines represent non-linear regression curve fit based on the averaged raw data points plotted using GraphPad Prism 10.0. **f** Initial kinetic rate constants ($k_{cat}$ (s$^{-1}$), $K_m$ (µM), $k_{cat}/K_m$ (s$^{-1}$µM$^{-1}$) of CXE15 and CXE20. **g, h** Electrostatic surface representation of AtCXE15 and AtCXE20 (left). Zoom in view of CXE15 (**g**) and CXE20 (**h**) catalytic pockets (middle), and sequence conservation of selected residues (right) within the corresponding catalytic pocket. The electrostatic potential is calculated by PyMOL and APBS with the non-linear Poisson−Boltzmann equation contoured at ±5 kT/*e*. Negative and positively charged surface areas are colored in red and blue, respectively. Naming scheme, At *A. thaliana*, Bn *Brassica napus*, Cr *Caspella rubella*, Cs *Camelina sativa*, Tc *Theobroma cacao*, Gh *Gossypium hirsutum*, Ma *Mercurialis annula*, Ah *Arachis hypohaea*, Cp *Carcia papaya*, Rc *Ricinus communis*, Cb *Capsicum baccatum*, Mc *Momordica charantia*, Tw *Tripterygium wilfordii*, Pa *Populus alba*, Dz *Durio zibethinus*. All experiments in (**a−f**) were repeated three times. Source data are provided as a Source Data file.

compact cavity compared to CXE20 which has a deeper and wider cleft (Fig. 2g, h). The main difference between the pockets can be attributed to the presence of negatively charged and hydrophobic residues in CXE15, and positively charged and hydrophobic residues in CXE20 (Fig. 2g, h). Together, the structural differences within the catalytic core can offer a partial explanation for the contrasting enzymatic activities of CXE15 when compared to CXE20.

**The NT helix of CXE15 plays a critical role in SL catabolism by facilitating structural plasticity for optimal enzymatic activity**

Given the distinct differences in activity and the probability of structural plasticity in the core and NTH region, we employed comparative molecular dynamics (MD) simulations for CXE15 and CXE20 using the crystal structures described above, both in the presence and absence of SL analog (*rac*-GR24). Root mean square deviation (RMSD) was

evaluated as a function of time spanning a duration of 200 nanoseconds (ns). No significant variation in the RMSD values was detected between CXE15 (*apo* form) and CXE15 bound to SL (CXE15-SL), reaching a characteristic plateau after 10 ns, and stabilizing within a range of 0.1–0.2 Å (Fig. 3a). RMSD values for CXE20 and CXE20-SL displayed a similar plateau after 10 ns (Fig. 3b), indicating that the systems remained stable throughout the MD simulations for both CXE proteins. Next, we monitored the structural flexibility of CXE15 and CXE20 by calculating the root mean square fluctuation (RMSF). The RMSF analysis of CXE20 did not reveal a similar movement of its NT region (Fig. 3b). The SL binding pocket loop connecting α7–α8 of CXE20-showed reduced dynamics in the presence of SL (Fig. 3b and Supplementary Fig. 7a) and the insertion of Y13 from the NT region of CXE20 into the cavity, creates a kink in the binding pocket loop to prevent steric clashes (Supplementary Fig. 7a). Notably, the presence of F31 and R307 in CXE20 may hinder the proper orientation of the ABC-ring of SLs and their dynamics resulted in larger pocket size (from 303 $A^2$ to 620 $A^2$, Supplementary Fig. 7a, b), and can explain reduced enzymatic activity. Interestingly, in comparison to CXE15-*apo*, our MD simulation analysis of CXE15-SL exhibited higher flexibility centered in the loops connecting the NTH to the catalytic core, as well as the loop (residue positions 215–222) that reside in proximity to the binding pocket (we termed Binding-Pocket Loop, BPL) (Supplementary Fig. 7c). Strikingly, CXE15 features N307, a smaller side chain group in place of R307, and the loop connecting the NT to the catalytic core extends farther away from the core compared to CXE20 (Supplementary Fig. 7d). The BPL exhibits two distinct states between the *apo* and CXE15-SL structures and seems essential for fine-tuning the correct positioning of the ABC-ring of SL (Supplementary Fig. 7c–e). Notably, our analysis revealed a major transition of the NTH from an open (*apo*) to a closed state (CXE15-SL) acting as a lid covering the catalytic pocket (Fig. 3a, Supplementary Fig. 7c–e, and Supplementary Movie 1). It is highly likely, therefore, that the NTH plasticity facilitates the robust catalytic process.

To further address the role of the NTH of CXE15 in SL catalysis, we generated a truncated CXE15 without the NTH (CXE15$^{\Delta NTH}$). DSF analysis showed a moderate thermal shift ($\Delta T_m = -1.0$ °C, Fig. 3c) between CXE15$^{\Delta NTH}$ and CXE15 indicating slight destabilization of the NTH truncated mutant. Notably the presence of *rac*-GR24 resulted in relative stabilization of CXE15 ($\Delta T_m = +1$ °C, Fig. 3c), but the absence of the NTH (CXE15$^{\Delta NTH}$) leads to no significant thermal shift compared to CXE15 and slight stabilization when comparing with CXE15$^{\Delta NTH}$ without *rac*-GR24. This indicates that NTH truncation may allow some SL binding probably due to increased accessibility to the pocket. Next, we tested enzymatic activity of CXE15 and CXE15$^{\Delta NTH}$ using the YLG hydrolysis assay (Fig. 3d), as well as their activity in planta (Fig. 3e). Although CXE15$^{\Delta NTH}$ exhibited some hydrolytic activity, its catalytic efficiency was drastically reduced to one-quarter of that of CXE15 due to the absence of the NTH (Fig. 3d, f). Interestingly, in a similar reaction, the addition of the NTH peptide of CXE15 at increasing concentrations significantly restored the hydrolytic activity, underlining the requirement of the NTH (Fig. 3d).

Next, we tested the role of NTH in planta by transiently over-expressing *AtCXE15* and its variants in *Nicotiana benthamiana* plants using the potato virus X (PVX) expression system. Remarkably, over-expression of *CXE15* (*CXE15-OE*) resulted in axillary shoot outgrowth compared to overexpression of citrine as control (Fig. 3e and Supplementary Fig. 8a, b). Transient expression of *CXE20* resulted in no axillary shoot outgrowth, probably due to the lack of effective SL hydrolysis. Transient overexpression *CXE15* without NTH (*CXE15$^{\Delta NTH}$*) and *CXE15$^{S169A/E271A}$* catalytic mutant exhibited no branching phenotype compared to full-length *CXE15* (Fig. 3e and Supplementary Fig. 8a, b). These results were further supported by the reduced expression levels of the *BRC1* in the axillary buds of CXE15-OE plants compared to *CXE15$^{S169A/E271A}$* catalytic mutant (Supplementary Fig. 8c). The observed

branching phenotype, along with the distinct expression pattern of *BRC1*, as seen in other SL-deficient mutants[11,33], indicates a significant local depletion of endogenous SL due to the activity of the CXE15 enzyme. Together, the structural, biochemical, and planta results highlight the importance of NTH in regulating the CXE15 enzymatic function to effectively catabolize SL.

To further elucidate the function of NT region of CXE15 and CXE20, we generated NT swap mutants CXE20$^{CXE15-NTH}$ (NT-α/β of CXE20 was replaced with the NTH of CXE15) and CXE15$^{CXE20-NT\alpha/\beta}$ (NTH of CXE15 was replaced with the NT-α/β of CXE20) proteins. The catalytic activity of the CXE15$^{CXE20-NT\alpha/\beta}$ mutant was drastically reduced when compared to CXE15$^{\Delta NTH}$ (Fig. 3fi,ii), suggesting yet again the importance of both the CXE15 core and its NTH for robust activity and catalytic efficiency. The mutant CXE20$^{CXE15-NTH}$ was relatively inactive when compared to CXE20 which exhibited minimal catalytic activity (Fig. 3fi-ii). No significant changes were recorded for NT swap mutants via our DSF analysis in the presence or absence of *rac*-GR24, probably due to compromised activity of those swapped mutants (Supplementary Fig. 9a–c). Interestingly, the calculated $K_m$ showed a greater binding affinity for YLG by CXE20$^{CXE15-NTH}$ (7.5 µM ± 1.1) compared to the CXE15$^{CXE20-NT\alpha/\beta}$ (19.6 µM ± 2.5) and CXE15$^{\Delta NTH}$ (62.0 µM ± 12.6) mutants and the wildtype CXEs (values) tested (Fig. 3fii and Supplementary Fig. 9c). The CXE15$^{CXE20-NT\alpha/\beta}$ swap mutant exhibited slightly higher catalytic efficiency ($1.29 \times 10^{-2}$ s$^{-1}$ µM$^{-1}$) compared to the truncated mutant CXE15$^{\Delta NTH}$ ($1.03 \times 10^{-2}$ s$^{-1}$ µM$^{-1}$) (Fig. 3fii).

## The crystal structure of CXE15 trapped with a hydrolyzed intermediate provides insights into the SL catalysis mechanism

To gain deeper insights into the mechanism of SL catabolism, we determine the crystal structure of CXE15 treated with *rac*-GR24 (termed CXE15$^{GR24}$, Table 1 and Fig. 4) at 2.3 Å resolution. Inspection of the electron density revealed the presence of a hydrolyzed lactone D-ring of *rac*-GR24 covalently linked to S169 of the catalytic triad (Fig. 4a–d). While the overall structure of CXE15$^{GR24}$ is highly similar to its CXE15 *apo* state (Fig. 4c), there are distinct differences in the catalytic pocket (Fig. 4c–e). Compared to CXE15$^{apo}$, the catalytic pocket of CXE15$^{GR24}$ increases by 12% surface area (-276 Å$^2$) and 30% in volume (-209 Å$^3$, Fig. 4e). The carboxylate group of the D-ring, apart from being covalently linked to S169, is also stabilized by the backbone nitrogen atoms of residues in positions 85–86 (Gly-Gly), forming the oxyanion hole located above the catalytic S169 (Fig. 4b). In CXE15$^{GR24}$, the calculated distance between H302 and E271 (from 2.5 Å to 2.7 Å) and between S169 and H302 (from 3.6 Å to 3.8 Å) increased compared to CXE15$^{apo}$ (Figs. 4b, d). These changes are likely attributed to the formation of the acyl-enzyme complex and the release of a modified ABC-ring of GR24 (Fig. 4d–f). The D-ring is then stabilized by aromatic and hydrophobic residues, such as F203, F204, F229, and L221 (Fig. 4b, d).

Given the crystal structures of the *apo* and the intermediate bound state of CXE15, the mechanism for SL catalysis is proposed (Fig. 4f). CXE15 catalyzes SLs through a three-step process: (i) the formation an acyl-enzyme tetrahedral intermediate resulting from the nucleophilic attack (by S169 and the relay charge network of H302 and E271) on the carbonyl group of the D-ring ester bond; the resulting tetrahedral intermediate state is stabilized by the oxyanion hole; (ii) in the acid-catalyzed process, the leaving group picks up the proton from H302 and this results in the release of ABC ring of GR24 as an alcohol component (R1–OH) while the D-ring is still intact with S169 and continued to be stabilized through hydrogen bonds formed by glycine residues in the oxyanion pocket; (iii) a water molecule (glutamic acid can act as a base and assist in the deprotonation of proximate water molecule, thereby making the water nucleophilic) attacks the carbon at C2′ position resulting in further catalysis resulting in the release of modified D-ring. This restores the catalytic residues to their initial positions for the commencement of a new catalytic cycle. Next, we

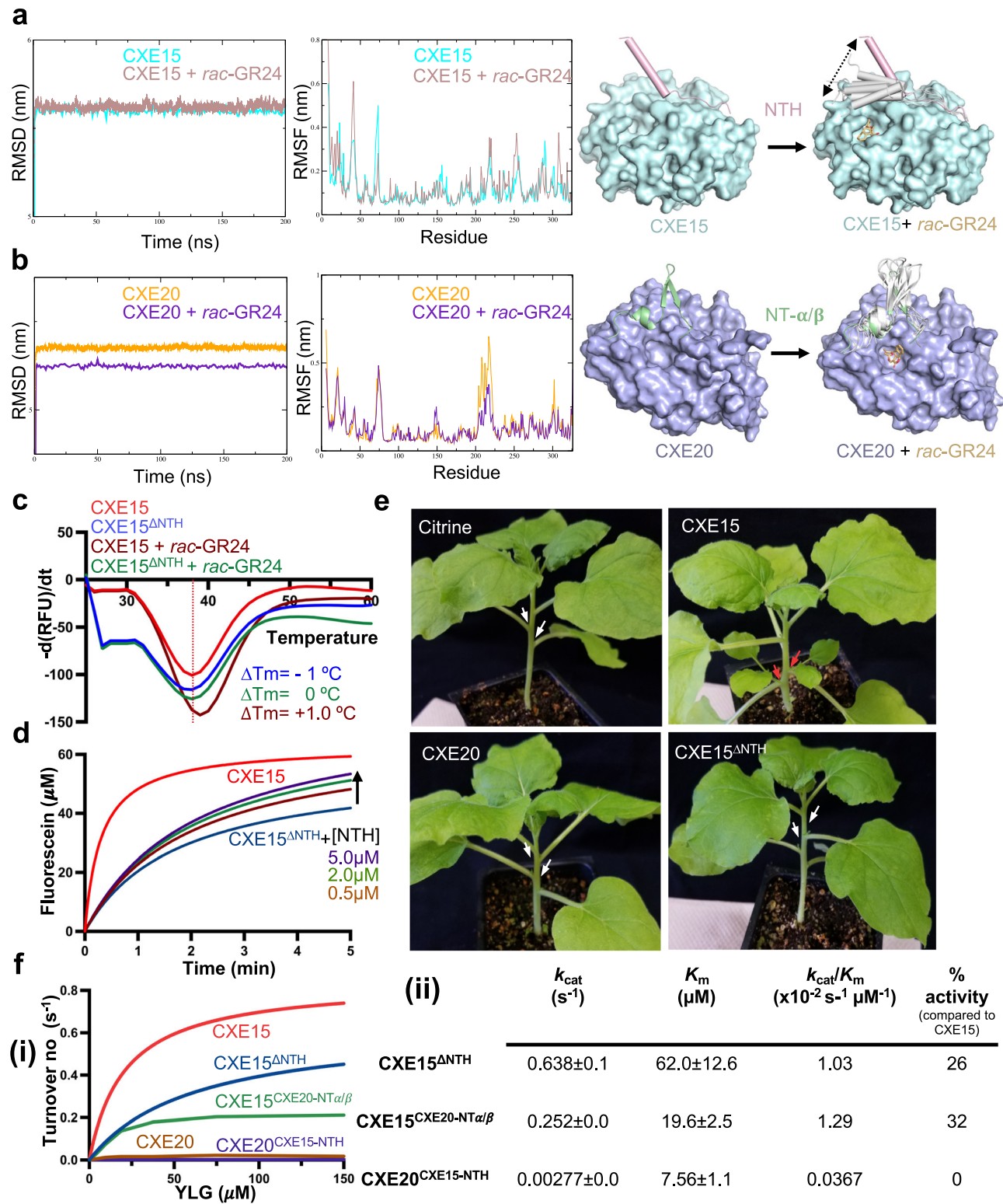

**Fig. 3 | NTH plasticity of AtCXE15 facilitates SL catabolism. a, b** MD simulation of apo-CXE15 (cyan), CXE15$^{rac\text{-}GR24}$ complex (brown), apo-CXE20 (orange), and CXE20$^{rac\text{-}GR24}$ complex (purple). RMSD of all atoms of CXE15, CXE15$^{rac\text{-}GR24}$ (top-left) and CXE20, CXE20$^{rac\text{-}GR24}$ (bottom-left) plotted as a function of time. RMSF of all atoms of CXE15, CXE15$^{rac\text{-}GR24}$ (top-middle), and CXE20, CXE20$^{rac\text{-}GR24}$ (bottom-middle) plotted as a function of residue position. The structure from the trajectories is superimposed to analyze the dynamic nature of the NT region of the CXE15$^{rac\text{-}GR24}$ (cyan) and CXE20$^{rac\text{-}GR24}$ (purple) complex (right). **c** Melting temperature curves of CXE15 (red), CXE15 with *rac*-GR24 (brown), CXE15$^{\Delta NTH}$ (blue), and CXE15$^{\Delta NTH}$ with *rac*-GR24 (green) were determined through DSF. **d** YLG hydrolysis assay of CXE15

(red), CXE15$^{\Delta NTH}$ (blue) titrated with 0.5 μM (brown), 2.0 μM (green), and 5.0 μM (purple) concentration of NTH peptide. **e** Agrobacterium with PVX-Citrine, PVX-CXE15, PVX-CXE20, and CXE15$^{\Delta NTH}$ were infiltrated into two opposite leaves of 3-weeks *N. benthamiana* plants. Representative photographs shown were taken two weeks post-infiltration. Experiments were repeated twice. Red arrows point to auxiliary branches and white arrows point to no auxiliary branches. **f** YLG hydrolysis assay (i) and initial rate kinetic constants (ii) of CXE15 (red), CXE15$^{\Delta NTH}$ (blue), CXE20 (brown), CXE15$^{CXE20\text{-}NT\alpha/\beta}$ (green) and CXE20$^{CXE15\text{-}NTH}$ (purple). Source data are provided as a Source Data file.

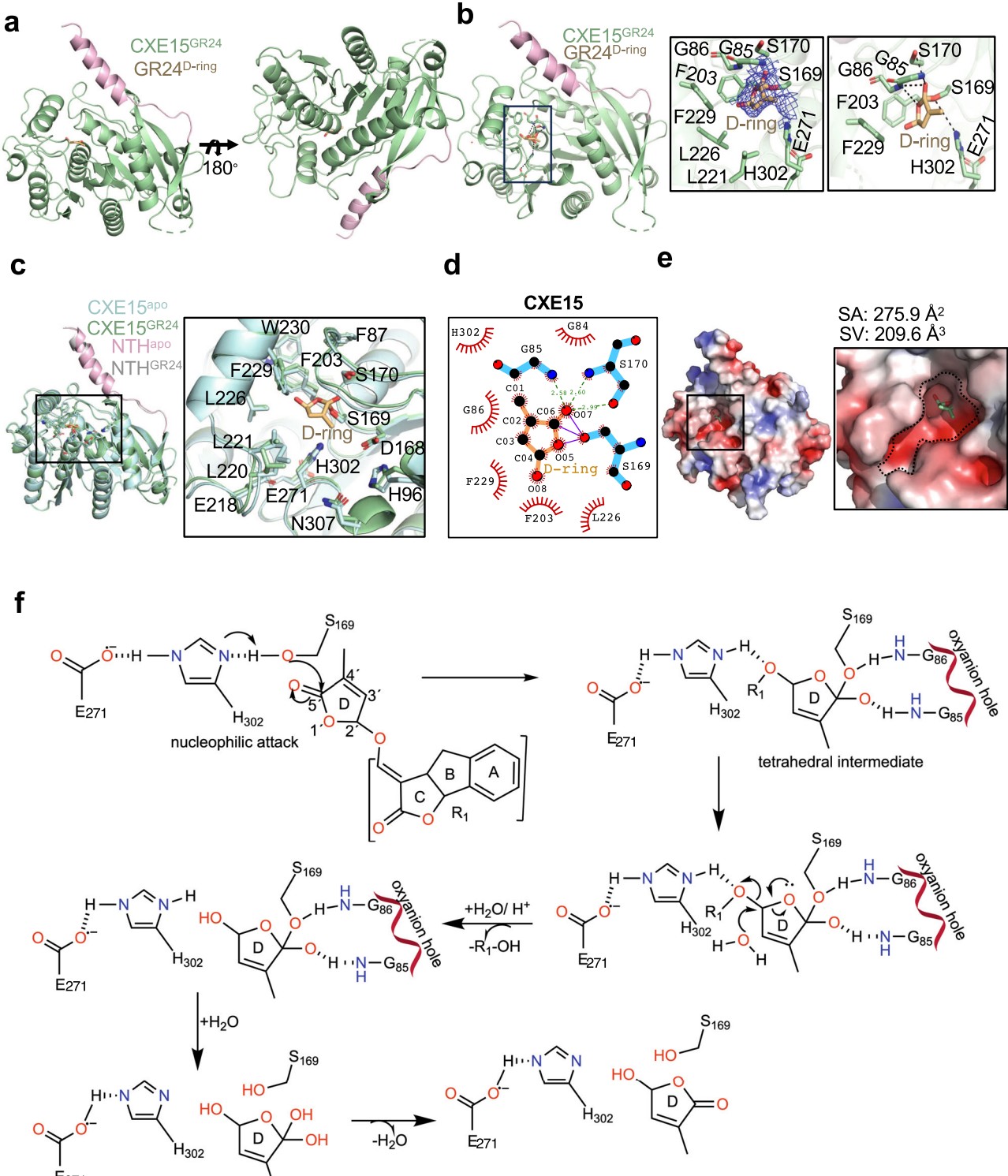

examined the catalysis by incubating CXE15 and CXE20 with *rac*-GR24 and recording mass spectrometry (MS) spectra (Supplementary Fig. 10) under denaturing conditions. As expected, no significant shifts in mass were recorded for CXE15 (Supplementary Fig. 10a), likely due to its rapid hydrolytic activity in solution and the fast release of the products impeding detection, as previously shown for highly efficient SL hydrolysis by certain KAI2s[48]. The recorded mass spectra for CXE20 revealed a notable mass shift corresponding to the D-ring of GR24 (Supplementary Fig. 10b). Subsequent trypsin digestion and nano-LC-MS/MS analysis identified the presence of a 96 Da adduct on the

catalytic H302 of CXE20. This mechanism of CXE20, as proposed in Supplementary Fig. 11, aligns with the slow rate SL catalysis by certain receptor-hydrolases, D14/KAI2, wherein a covalent adduct of the D-ring to the catalytic H247 of D14s results in the release of the ABC-tricyclic ring of GR24[48,49].

## Discussion

The increasing number of studies exploring enzymes involved in phytohormone post-signaling deactivation has undoubtedly expanded our understanding of these processes[24,28,50–52]. However,

**Fig. 4 | Crystal structure of AtCXE15$^{GR24}$ complex and structural investigation of SL binding and catalysis. a** Crystal structure of CXE15$^{GR24}$ (pale green) with distinct NTH (light pink) shown in different angles. **b** Overall structure of CXE15$^{GR24}$ bound to D-ring (sticks, light orange) of GR24 (left). Close-up view of D-ring interaction within the catalytic cavity. The $F_o$–$F_c$ map of the D-ring is shown at 1σ cut-off (middle-right). The γ-oxygen of the catalytic S169 of CXE15 is covalently linked to the carboxylate group of the D-ring. **c** Comparative structural analysis of CXE15$^{apo}$ (pale cyan) and CXE15$^{GR24}$ (light green) complex (left). Close-up view of catalytic pocket showing the differences in the orientation of amino-acid side chains when bound to D-ring of GR24 (right). **d** Ligplot interaction diagram of CXE15$^{GR24}$ bound to D-ring (light orange). The covalently linked residues are shown as straight lines, the hydrogen bonds are represented as dashed lines and the arc represents the hydrophobic interaction. **e** Electrostatic representation of CXE15 bound to D-ring (left). Zoom in view into the catalytic pocket of CXE15$^{GR24}$ bound D-ring (sticks).

Dashed lines indicate the catalytic cavity. The electrostatic potential is calculated by PyMOL and APBS with the non-linear Poisson–Boltzmann equation contoured at ±5 $k$T/$e$. Negative and positively charged surface areas are colored in red and blue, respectively. **f** Proposed schematic representation of the mechanism of action of CXE15 towards SLs in plants. S169 is deprotonated by histidine (H302) which is stabilized in its charged form by the presence of Glutamic acid (E271). This deprotonated S169 then undergoes a nucleophilic attack on the carbonyl of the substrate (GR24). The resulting tetrahedral transition state is stabilized by an oxyanion hole (G85–G86). A water molecule subsequently performs a nucleophilic attack, protonating the leaving group (ABC-ring of GR24) and forming a covalent enzyme-substrate intermediate. In the second phase of the reaction, another water molecule attacks the covalent intermediate, resulting in the formation of the product and the restoration of the catalytic triad to its native state. Source data are provided as a Source Data file.

significant gaps persist in our knowledge concerning the identification, biochemical characterization, and regulation of these enzymes within signaling pathways. Recent investigations in the SL signaling pathway revealed the possible role of carboxylesterases, specifically CXE15 and CXE20 in *Arabidopsis*. Yet it remained unclear what is the precise mode of action of these CXEs in SL catabolism. In this study, we analyzed the plant carboxylesterases through phylogenetic examination, uncovering among other critical motifs (such as HGGG and GXSXG that play a role in catalysis), a distinct yet highly conserved NT region across various CXE clades. The divergence in the NT sequences, despite the conservation of key residues and core domains, suggests that this region may have evolved to serve specific functions in different CXE clades. Nevertheless, further investigation is warranted to uncover the precise role of the NT region in the many other non-SL-related carboxylesterases.

We determined at high resolutions the crystal structures of CXE15 (*apo* and bound) and CXE20 and uncovered the distinct NT domains between CXE15 (NTH) and CXE20 (NT α/β fold). We found that there is a dynamic transition of the NTH domain in CXE15 from an open to a closed state and these structural differences were also associated with changes in the size and shape of the substrate binding pockets of these enzymes. This modulation facilitates robust SL hydrolysis by CXE15, highlighting a unique regulatory mechanism not observed in CXE20.

CXE enzymes represent a highly conserved superfamily of enzymes from bacteria to humans. It has been previously suggested for several esterase and lipase enzymes that changes in the binding pocket dimension can impact specificity and catalytic efficiency[53–55]. Indeed, the larger dimension of the plant CXE20 SL binding pocket suggests additional plasticity, as it provides free space that can result in larger fluctuations upon SL binding. This, in turn, may lead to reduced catalytic specificity due to suboptimal SL fitting. On the other hand, CXE15 exhibits a compact catalytic pocket shape, which is likely to enhance substrate specificity through specialized residues engaging with the substrate with the important contribution of NTH plasticity. It should be noted that our MD simulation may not capture all conformations under physiological conditions with the substrate, and other conformations may be triggered by the perception and catalysis process.

Our biochemical and planta analyses provided further insights into the functional consequences of the structural distinctions and the requirement of both the catalytic core domain and the NTH of CXE15. Notably, AtCXE15 demonstrated significantly higher enzymatic activity towards SL analogs, compared to AtCXE20, and the structure of AtCXE15 bound with SL analog, *rac*-GR24, intermediate provided important insights into a possible mode of action of SL binding and catalysis by CXE15. Our models suggest that plant CXE15 is more sensitive to SL when the NTH is in an open conformation and upon SL binding, the NTH transitions from an open to closed state (Fig. 5a). The open-close-open transition facilitates the repositioning of the binding

pocket loop, allowing the fine-tuning of the SL orientation for efficient and rapid breakdown within the catalytic pocket, followed by the quick release of the product(s). Plant CXE20 is likely to undergo larger conformational changes centered around the catalytic pocket upon SL binding, but unlike CXE15 its NT region is more rigid. The overall increase in the pocket width of CXE20, which is already wider than *apo*-CXE15, enables SL binding but could compromise efficient catalysis, likely due to the increased distance between the catalytic residues (Fig. 5b). Collectively, the in vitro, and the MS data further corroborate the slow catalysis of CXE20 by recording a post-hydrolysis covalent adduct on the catalytic H302, as observed in other early studies on slow hydrolyzing SL receptors[48,49].

Remarkably, *CXE15-OE* (compared to truncated or catalytically inactive CXE15) in *N. benthamiana* resulted in axillary bud outgrowth with a distinct expression pattern of *BRC1*, similar to other SL-deficient mutants. While this data strongly supports the biochemical function and structure observed here, it raises intriguing questions about the potential transport of the transiently expressed enzyme to the bud and/or the transport of SL to the axillary bud. Additionally, we cannot rule out the possibility that CXE15 may act on another molecule and/or SL precursor to promote axillary bud growth. Therefore, future studies combining endogenous SL detection with CXEs overexpression or knockout lines will clarify how these enzymes precisely fine-tune SL levels in planta.

Overall, this study contributes to our comprehension of the regulation of the SL signaling pathway through post-signal deactivation. Changes in SL biosynthesis and/or signaling pathways lead to modifications in plant architecture, such as shoot branching. Consequently, the modulation of SL levels, as well as homeostasis or signaling, has become a significant focus in crop systems, including plant breeding endeavors[56–59]. Taking a broader perspective, the carboxylesterase superfamily plays fundamental roles in numerous essential metabolic pathways across all kingdoms of life. We unveil a molecular architecture and intrinsic dynamics for carboxylesterases that have not been observed before, thereby adding an important layer of complexity to the mode of action of these crucial enzymes.

## Methods

### Phylogenetic and sequence conservation analyses

The 20 CXE genes from *Arabidopsis* were retrieved manually from the uniport database (Supplementary Data 1). The CXE15 and CXE20 homolog sequences were identified by BLASTp NCBI using *Arabidopsis* CXE15 and CXE20 as the query sequence (Supplementary Data 2 and Supplementary Data 3). The alignment of sequences was performed by MEGA11 using MUSCLE multiple alignment[60]. The phylogenetic tree was constructed by the Maximum Likelihood method[61] and colored with iDOL V5[62]. The NT region of CXE15 and CXE20 (Supplementary Data 4) were aligned by CLC Genomics Workbench v22 and the sequence logos were generated by WebLogo[63].

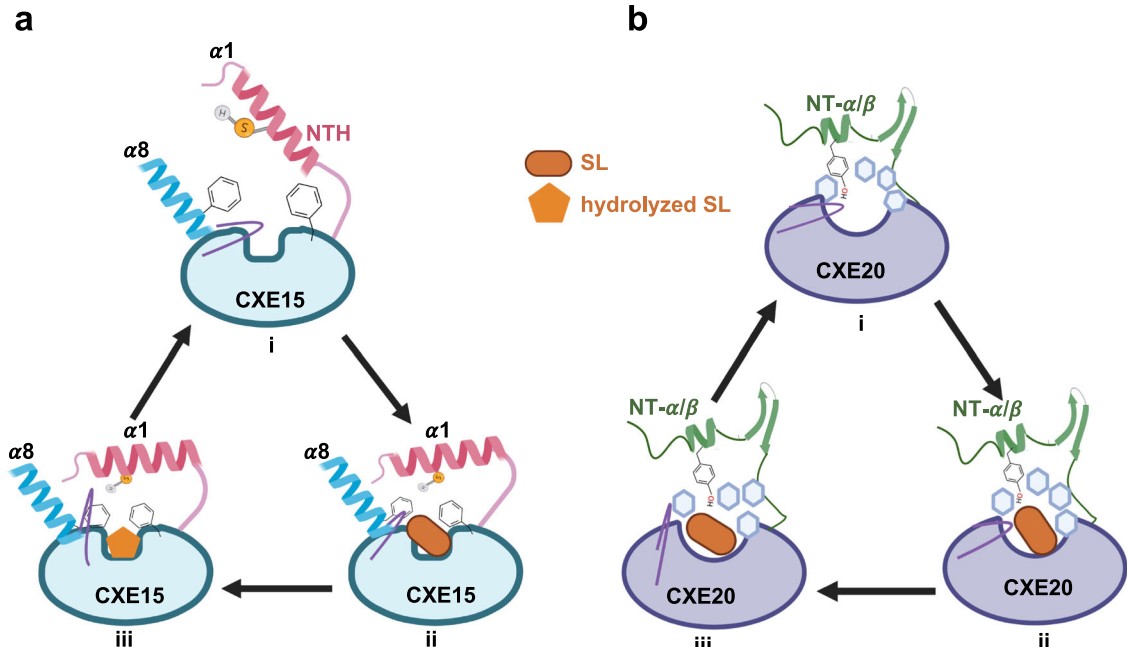

**Fig. 5 | Proposed modes of action of CXE15 and CXE20. a** (i) The NTH (light pink) of CXE15 adopts an open conformation, allowing access for SLs and solvent molecules to enter the catalytic cavity. The loop connecting α7–α8 sits directly above the cavity. Two phenylalanine(s), F89 and F229, extend on either side of the pocket above the catalytic cavity. (ii) In the presence of the hormone, SL (GR24, rectangle, orange), inside the cavity, the NTH closes the pocket. This conformational change results in the reorientation of F89 from an upward to a downward position towards the cavity. Additionally, the loop connecting α7–α8 transitions from a downward to an upward movement, enlarging the pocket's overall size, and providing more space for GR24 binding. Consequently, M15 from NTH becomes trapped by F89 and F229 through SH–C interactions on each side of the cavity until SL is hydrolyzed. (iii) The SL analog, GR24, undergoes enzymatic action by the catalytic residues S169, H302, and E271, causing the ABC ring's dissociation from the D-ring. The D-ring (five-membered ring, orange) forms a covalent linkage with S169 and is later modified by the action of CXE15. The fate of the D-ring, whether it

undergoes further enzymatic action by CXE15 and the resulting final product, remains unknown and requires further investigation. **b** (i) The NT α/β segments of CXE20 are adjacent to the catalytic pocket, and the amino acid Y13 extends toward the catalytic cavity. The catalytic cavity of CXE20 is encircled by a ring of phenylalanine(s), and the loop connecting α8–α9 points inward towards the cavity. Unlike CXE15, the pocket of CXE20 is wide open and spacious. (ii) In the presence of the SL, the loop connecting α8–α9 shifts from a downward to an upward position, leading to an increase in the overall size and shape of the pocket. The ring of phenylalanine(s) stabilizes the SL analog GR24 along with Y13 from the N-terminus. (iii) The binding of SL into the cavity results in only very weak hydrolysis, catalyzed by the amino acids S166, H302, and D272 due to the overall increase in the catalytic pocket size. This Figure was created with BioRender.com and released under a Creative Commons Attribution-NonCommercial-NoDerivs 4.0 International license.

## Protein preparation and purification

AtCXE15 was PCR amplified from *Arabidopsis* cDNA and then recombined into a pAL vector (Addgene) to express as a 6xHis-SUMO fusion protein. ΔN45AtCXE15 was PCR amplified from the AtCXE15 plasmid. To obtain AtCXE15[S169A] and AtCXE15[E271A], site-directed mutagenesis was carried out using the restriction-free cloning protocol. AtCXE20, CXE15[CXE20-NTα/β], and CXE20[CXE15-NTH] were PCR amplified from codon-optimized synthesized gene from Twist Bioscience. All these constructs were recombined into a pAL vector to express as a 6xHis-SUMO fusion protein. The sequence confirmed plasmids were transformed into *Escherichia coli* strain BL21 (DE3) for expression and purification.

AtCXE15, AtCXE15[S169A], AtCXE15[E271A], AtCXE20, AtCXE15[ΔNTH], AtCXE15[CXE20-NTα/β], and AtCXE20[CXE15-NTH] were independently cloned and expressed as a 6×His-SUMO fusion protein from the expression vector pAL. BL21 (DE3) cells transformed with the expression plasmid, grown in LB broth at 16 °C to an $OD_{600}$ of ~0.8, and induced with 0.3 mM IPTG for 16 h. Cells were harvested, re-suspended, and lysed in extraction buffer (50 mM Tris, pH 8.0, 200 mM NaCl, 5 mM imidazole, and 1 mM tris(2-carboxyethyl) phosphine (TCEP)). All these constructs were expressed and isolated from soluble cell lysate by Ni-NTA resin eluted with 250 mM imidazole and subjected to anion exchange. The eluted protein was then cleaved with TEV (tobacco etch virus) protease overnight at 4 °C. The cleaved His-SUMO tag was removed by passing through a Nickel Sepharose and further purified by chromatography through a Superdex-200 gel filtration column in 20 mM HEPES, pH 7.2, 150 mM NaCl, and 5 mM DTT. All proteins were concentrated by

ultrafiltration to 8–10 mg/mL⁻¹. All uncropped gels and scans are provided in Supplementary Fig. 12.

## Plant growth conditions and transient expression in *N. benthamiana*

*N. benthamiana* seeds were sown on Sunshine mix 1 and were grown in a controlled environment chamber with 14 h light and 10 h dark photoperiod at 24 °C. Coding sequences of CXE15, CXE20, CXE15[ΔNTH], CXE15[S169A/E271A], and citrine were cloned into PVX expression vector SPDK658[64]. These PVX derivatives were transformed into Agrobacterium strain GV2260. Agrobacterium cultures ($OD_{600}$ = 0.5) containing various PVX vectors were infiltrated into two lower leaves of the four-leaf stage, 3-week-old *N. benthamiana* plants using a 1-mL needleless syringe. Infiltrated leaves were monitored for four weeks and photographs of whole plants were taken periodically.

## Quantitative real-time PCR

To assess the *NbBRC1* expression level, axillary bud tissue collected from PVX-infected plants was used to extract total RNA using TRIzol reagent (Life Technologies) by following the manufacturer's protocol. Total RNA was treated with DNase I RNase free (New England Biolabs). The first-strand cDNA was generated from 1 μg of total RNA using the oligo(dT) primer and Superscript III Reverse Transcriptase (Thermo-Fisher Scientific). qPCR was performed using a Bio-Rad CFX96 Touch Real-Time PCR detection system (Bio-Rad) using iTaq universal SYBR Green Supermix (Bio-Rad) and NbBRC1 specific primers (NbBRC1-F: 5′-

TGGTGCAATTAGTACTGCAATATC-3'; NbBRC1-R: 5'- TCCTTTAGCGG TTTCCAGCTTC-3'). NbeIF4A with primers (NbeIF4A-F: 5'- GCTTTGG TCTTGGCACCTACTC and NbeIF4A-R: TGCTCGCATGACCTTTTCAA) was used as an internal control to normalize the data. The fold change in mRNA levels was determined using the ΔΔCt method.

## DSF

Samples were prepared in triplicate using 10 μM protein, incubated with 500 μM *rac*-GR24 in a buffer containing 20 mM HEPES, 150 mM NaCl, and 1 mM TCEP. Following 20 min of incubation with the *rac*-GR24, 0.5 mM Sypro Orange was added and used as the reporter dye. Following a 5-min equilibration at 4 °C, samples were heat denatured using a linear 4–95 °C gradient at a rate of 1 °C min$^{-1}$. Protein unfolding was monitored by detecting changes in Sypro Orange fluorescence using a Bio-Rad CFX96 real-time system with HEX emission (533 nm) and excitation (559 nm) fluorophore settings.

## YLG activity assays

YLG (ThermoFisher Scientific) hydrolysis assays were conducted in 50 μL of reaction buffer (50 mM MES pH 6.0, 250 mM NaCl, and 1 mM dithiothreitol) in a 96-well, F-bottom, black plate (Greiner) according to manufacturer's instructions. The intensity of the fluorescence was measured using a Synergy Microplate Reader (BioTek) with excitation at 480 nm and detection at 520 nm. Readings were collected at 2 s intervals over 2 min. The background signal of YLG is subtracted from the protein signal and later converted to Fluorescein per micromolar. The rate of YLG hydrolysis was determined from the initial linear portion of this fluorescence data. Kinetic constants were determined through non-linear regression based on Michaelis Menten kinetics using Prism Graphpad v.10. In all experiments the CXE enzyme, at 1 μM, for both CXE15 and CXE20, 2 μM for CXE20$^{CXE15-NTH}$ and CXE15$^{\Delta NTH}$, and 5 μM for CXE15$^{CXE20-NT\alpha/\beta}$; was added to a reaction mixture containing 0–150 μM YLG and shaken for 5 s before data was collected. NTH peptide: MGSLGEEPQVAEDCMGLLQLLSNGTVLRSE was synthesized (Biometik) and dissolved in DMSO to a final concentration of 5 mM. In assays the peptide was diluted in reaction buffer and equivalent volumes of DMSO were added as to all experimental tubes.

## Trypsin-limited proteolytic digestion

Purified AtCXE15 and AtCXE20 (1 mg/mL) were incubated at 22 °C in the presence and absence of *rac*-GR24 for 15 min with trypsin solution containing 1 μg/mL trypsin (Goldbio T-160) at 50 mM MES pH 6.5, and 1 mM dithiothreitol. The proteolysis reactions were stopped by a fourfold concentration of SDS-PAGE sample buffer, followed immediately by 5 min boiling at 95 °C. Proteins were resolved by SDS-PAGE and Coomassie stain. All uncropped gels and scans are provided in Supplementary Fig. 12.

## Crystallization, data collection, and structure determination

Crystals of AtCXE15 were grown at 22 °C by the hanging-drop vapor diffusion method with 2.0 μL of protein sample mixed with 1.0 μL of a reservoir solution containing 0.2 M sodium/potassium phosphate pH 7.5, 0.1 M HEPES pH 7.5, 22.5% *v/v* PEG smear medium, 10% glycerol. Crystals of maximal sizes were obtained after 1 week and flash-frozen in liquid nitrogen with 20% glycerol as cryoprotectant. To obtain the AtCXE15$^{GR24}$ complex, crystals obtained from the above condition were soaked in the mother liquor containing 1 mM of *rac*-GR24 at different time intervals and flash-frozen in liquid nitrogen using 20% ethylene glycol as cryoprotectant. Crystals of AtCXE20 were grown at 22 °C by the hanging-drop vapor diffusion method with 2.0 μL of protein sample mixed with 1.0 μL of a reservoir solution containing 10% *w/v* PEG8000, 20% *v/v* ethylene glycol, 0.1 M imidazole, 0.03 M sodium nitrate, 0.03 M disodium hydrogen phosphate, and 0.03 M ammonium sulfate. Crystals of maximal sizes were obtained after 4 days, and flash frozen in liquid nitrogen with 20% glycerol as cryoprotectant. X-ray diffraction datasets were collected, integrated, and scaled using the HKL2000 package. Initially, the AlphaFold models of CXE15 and CXE20 were employed as search models to obtain solutions via phaser. However, the solution obtained using the AlphaFold model of CXE15 resulted in poor phasing. To overcome this issue, the first 50 amino acids were removed from the AlphaFold model, and the truncated structure was utilized as a search model, leading to a significant improvement in the solution quality with clear electron density observed for the initial 50 amino acids. Subsequently, the model-building feature in Coot was utilized to construct the initial 50 amino acids of CXE15. This model was further refined using Phenix.refine, resulting in final $R_{work}$ and $R_{free}$ values of 20% and 24%, respectively. In contrast to CXE15, the AlphaFold model of CXE20 provided a compatible structure solution. Consequently, the solution structure was refined further using Phenix.refine, resulting in final $R_{work}$ and $R_{free}$ values of 16.9% and 19.7%, respectively. All structural models were subsequently manually rebuilt and refined using COOT[65] and PHENIX[66].

## MD simulation

MD simulation of CXE15, CXE20, CXE15$^{GR24}$, and CXE20$^{GR24}$complexes were performed using GROMACS 2020.3 package[67,68] with CHARMM36 force field[69]. For the complex structures, the GR24 geometry files were generated using PRODRG server[70]. The initial structures were immersed in periodic equilibrated water boxes using the SPC (simple point charge) solvent model, forming cubic shapes with a distance of 1.2 nm between the solute to the box edge. Additionally, sodium ions were added to neutralize the system. Electrostatic interactions were computed using the particle mesh Ewald method[71] with a cutoff radius of 10 Å, and van der Waals interactions were truncated at 10 Å unless specified otherwise. The P-LINCS algorithm[72] was employed to maintain bond lengths at their equilibrium values. Following energy minimization, each system underwent equilibration for 1000 ps under NVT (constant number of particles, volume, and temperature) conditions, followed by a 1000 ps run in the NPT (constant number of particles, pressure, and temperature) ensemble with positional restraints applied to solute-solvent molecules with respect to protein-*rac*-GR24. The temperature was controlled at 300 K using Verlet-rescaling, and a Berendsen thermostat[68] was used to maintain pressure at 1 bar. The final MD simulations were conducted for 200 ns under the same conditions, except the position restraints were removed. Analysis of the results was performed using software from the GROMACS package, and graphs were generated using Xmgrace V5.1.25.

## Direct electrospray ionization−MS analysis of protein−ligand complex under denaturing conditions

After a 10 min incubation of CX20 and CX15 with (±)-GR24 500 μM, MS measurements were performed with an electrospray Triple-TOF 4600 mass spectrometer (ABSciex) coupled to the nanoRSLC ultra-performance liquid chromatography system (Thermo Scientific) equipped with a C4-desalting column. For ESI−MS measurements, the instrument was operated in positive and RF quadrupole modes with the TOF data being collected between *m/z* 400 and 2990. Collision energy was set to 10 eV and azote was used as collision gas. Mass spectra acquisition was performed after loading and desalting of protein samples on the C4 column. The Analyst and Peakview software were used for acquisition and data processing, respectively. Mass spectra were deconvoluted with the MaxEnt algorithm. The protein average masses are calculated from the spectra with a mass accuracy of ±1 Da.

## Localization of the fixation site of ligands on CXE20 and CXE15

MS measurements under denaturing conditions were carried out as described in de Saint Germain et al., 2021[73]. Localization of the fixation

site on AtCXE15, AtCXE20, AtCXE15-*rac*-GR24, and AtCXE20-*rac*-GR24 mixtures were incubated for 10 min prior to overnight trypsin lysis. Trypsin digested peptide mixtures were analyzed by nanoLC-MS/MS with the Triple-TOF 4600 mass spectrometer (AB Sciex) coupled to the nanoRSLC UPLC system (Thermo Fisher Scientific) equipped with a trap column (Acclaim PepMap 100 C18, 75 µm i.d. × 2 cm, 3 µm) and an analytical column (Acclaim PepMap RSLC C18, 75 µm i.d. × 25 cm, 2 µm, 100 Å). Peptides were loaded at 5 µL/min with 0.05% (*v/v*) TFA in 5% (*v/v*) acetonitrile and separated at a flow rate of 300 nL/min with a 5–35% solvent B [0.1% (*v/v*) formic acid in 100% acetonitrile] gradient in 40 min with solvent A [0.1% (*v/v*) formic acid in water]. NanoLC-MS/MS experiments were conducted in a Data-Dependent acquisition method by selecting the 20 most intense precursors for collision-induced dissociation fragmentation with the Q1 quadrupole set at a low resolution for better sensitivity. Raw data were processed with the MS Data Converter tool (AB Sciex) for generation of.mgf data files and proteins were identified with the MASCOT search engine (Matrix Science) against the AtCXE15 and AtCXE20 sequence with oxidation of methionine residues and ligand-histidine adduct as variable modifications. Peptide and fragment tolerance were set at 20 ppm and 0.05 Da, respectively. Only peptides were considered with a MASCOT ion score above the identity threshold[74] calculated at a 1% false discovery rate.

### Reporting summary

Further information on research design is available in the Nature Portfolio Reporting Summary linked to this article.

## Data availability

The atomic coordinates of AtCXE15, AtCXE15[GR24], and AtCXE20 structures generated in this study have been deposited in the Protein Data Bank with accession codes: 8VCA, 8VCD, and 8VCE, respectively. All data generated in this study are provided in the Supplementary Information, and Supplementary Data. Source data are provided in this paper.

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

## Acknowledgements

N.S. is supported by the National Science Foundation (NSF-CAREER Award #2047396, NSF- and Award #2139805), and by the U.S. Department of Energy, Office of Science, Biological and Environmental Research, Genomic Science Program grant no. DE-SC0023158. N.S. and S.P.D.-K. are supported by NSF-EAGER award #2028283. We thank the beamline staff at the Advanced Light Source (U.S. DOE Office of Science User Facility under contract no. DE-AC02-05CH11231, is supported in part by the ALS-ENABLE program funded by the National Institutes of Health, National Institute of General Medical Sciences, grant P30 GM124169-01).

## Author contributions

M.P. and N.S. conceived and designed the experiments. M.P. and L.Y. conducted the protein purification, biochemical, and crystallization experiments with the help of A.K.G. Structural, functional, and MD simulations analyses were determined and performed by M.P., and N.S. U.N. and S.P.D.-K. generated PVX vectors with different CXEs and performed in planta experiments. D.C. and F.D.B. performed the MS experiments. M.P. and N.S. wrote the manuscript with the help of A.K.G., L.Y., U.N., D.C., F.-D.B., and S.P.D.-K.

## Competing interests

The authors declare no competing interests.
