## [Peer Review File · Nature Communications]

REVIEWER COMMENTS

Reviewer #1 (Remarks to the Author):

Phytohormones levels are regulated through specialized enzymes, participating not only in their biosynthesis but also in post-signaling processes for signal inactivation and cue depletion. CXE15 and CXE20 carboxylesterases have been shown to deplete strigolactones (SLs) that coordinate various growth and developmental processes and also function as signaling molecules in the rhizosphere. In this study, the authors elucidate the crystal structures of CXE15 (both apo and SL intermediate bound) and CXE20 that revealed new insights into the mechanisms of SL perception and catabolism. The N-terminal region of CXE15 and CXE20 has distinct secondary structures that play pivotal roles in their function both in vitro and in planta. Their findings indicate that a dynamic transition of the open-closed N-terminal helix domain in CXE15 facilitates robust SL hydrolysis. The result not only illuminates the distinctive process of phytohormone breakdown but also uncovers a novel molecular architecture and mode of plasticity within a specific class of carboxylesterases. The manuscript is well written and the conclusion is reliable. However, I suggest the following minor corrections should be made before the manuscript can be accepted for publish.

1. The authors failed to get the AtCXE15 crystal structure resolved by the full length of CEX15 alphafold model, however, after deletion of a truncated (1-50aa) alphafold search model, they were successful in getting the crystal structure of full length CEX15. In the crystal structure of CEX15, a cap domain (residues 1-21) and a core catalytic domain (residues 32-329) had covered the residues 1-50. Since there is a deletion of residues 1-50, how can the authors resolved the crystal structure without these residues?
2. There were several methods to establish the model, such as selenomethionine-modified structure. Can author confirm the reliability of structure predicted by alphafold?
3. How about the similarity of AtCXE15 crystal structure compared to CEX crystal structure from other species?
4. There is a lack of the phenotypes for the mutagenesis of residues that are critical for GR24 binding in AtCXE15.
5. Are there any other methods to test the binding affinity between GR24 and AtCXE15/20 instead of DSF analysis and limited proteolytic digestion?
6. Supplementary Figure 3, the author should marked mAU with 280.
7. line 474, The initial structure was immersed in a periodic water box of cubic shape (1.0 nm). This solvent box is too small, and the result is not reliable.
8. line 480, We subsequently applied LINCS64 constraints for all bonds, keeping the whole protein molecule fixed and allowing only the water molecule to move to equilibrate with respect to the

protein structure. The procedures are not clear. Are there any restraints for the minimization and all equilibration steps?

Reviewer #2 (Remarks to the Author):

This manuscript describes the strigolactone (SL) catabolism mechanism by carboxylesterases based on those crystal structures. CXE15 and CXE20 have been proposed to catalyze SL hydrolysis, which is possibly the active hormone degradation step. The authors of this manuscript successfully solved the crystal structures of CXE15 and CXE20 at high resolution and proposed that C-terminal helix region in CXE15 has a critical role in its catalytic function. The results of this manuscript are clear and reliable. However, I think the results of this manuscript do not have enough general interest to attract the readers of this journal. The biochemical function of CXE15 as the SL hydrolyzing enzyme has been proposed in this manuscript and a previous report, however, I think the scientific evidence is still not enough to prove the involvement of this enzyme in the SL deactivation process in vivo. The authors mention that 'cxe15 mutants were more sensitive' by citing a previous report (Enjun Xu et al, Nat Plants, 2012). However, this result was obtained by a hypocotyl elongation assay. If considering that the main physiological role of SL is shoot branching regulation, such data is not enough to conclude the involvement of CXE15 in SL catabolism in vivo. Moreover, there were no data directly showing that endogenous SL levels are significantly decreased in the CXE15 overexpressing plants. I do understand that the structural characterization of these enzymes is quite valuable in the field, however, it is likely to be of interest only to people in the limited field. Some major and minor comments are described below.

Major comments

1. line; 74-75, CYP707A is not involved in the Jasmonoyl-L-isoleucine deactivation. The authors should correctly explain this part.

2. line 269-274. To conclude that the branching phenotype in the CXE15 expressing plant is due to SL degradation, the authors should analyze the endogenous levels of SLs.

3. line 308-322, The authors solved the crystal structure of the SL D-ring bound form of CXE15. Moreover, a unique reaction mechanism was proposed based on this intermediate structure. However, it is unlikely that the structural data is clear enough to get such a conclusion. To prove such a reaction mechanism proposed in Fig 4f, the authors should provide MS spec data which show that the entire GR24 molecule is covalently linked with CXE15. I do not understand how the water molecule is activated when it attacks the covalently linked intermediate.

4. As mentioned above, I am still skeptical that these enzymes contribute to the deactivation of SLs. If CXE15 is indeed involved in the SL catabolism, the *cxe15* mutant should accumulate high levels of SLs. Moreover, introduction of full-length CXE15, but not its C-terminal helix truncated form, should complement this phenotype. These data would be necessary to conclude the physiological function of CXE15.

Minor comments

1. The authors are using the words such as 'ligand' and 'perceive'. As far as I know these words are appropriate for explaining the receptor proteins. I think it would be better to use 'substrate' and 'bind/accomodate' instead.

2. line 157-161. The authors should explain the process of the CXE structural analysis more correctly.

Reviewer #3 (Remarks to the Author):

The manuscript by Palayam et al presents a very complete structural and biochemical study on two carboxylesterases (CXE15 and CXE20) from the model plant *Arabidopsis thaliana* and their role in catabolism of the phytohormone strigolactone. This is a well performed, experimentally rigorous, and clearly presented study that provides new insights on strigolactone homeostasis.

Major comments

1) In the title (and elsewhere in the manuscript), the authors use 'allosteric' regulation - this isn't accurate. Allostery implies an effector molecule that modulates (positive or negative) catalytic function at the active site. Perhaps 'conformational' regulation may more clearly fit what is observed in the x-ray crystal structures of CXE15 and CXE20.

2) At various points the abstract needs to provide more specific information for the reader. Line 25/26 - what organism? Perhaps add "*Arabidopsis thaliana*" before "CXE15 and CXE20 carboxylesterases ..." to let reader know what plant. Line 28 (and elsewhere in the text), "x-ray crystal structure" instead of just "crystal structure". The statement that the distinct secondary structures of the N-termini play pivotal roles for in vitro and in planta activity is a bit vague. What data supports this? Line 32 ... reword to "... transition of the N-terminal helix domain of CXE15 between open and closed forms facilitates ..." Line 34 - "novel" - overused word - replace or delete (same for other places in the manuscript text).

3) Results, line 157 - the fact that an AlphaFold model did not succeed in molecular replacement is not evidence (nor implies) a novel architecture. There are many reasons why a molecular replacement can fail. Consider removing some of this text.

4) In general, the experimental data provided provides new mechanistic insights on the CXE15 and CXE20 enzymes function. One question for the reviewers is the difference in N-terminal structure between the two enzymes. The movement of the helical cap of CXE15 is clear - the value of having apo and complexed versions of the structure is clear. Something for the authors to consider, are there other structural changes of the N-terminal region of CXE20 between open and closed forms - there are multiple examples of ligand binding triggering ordering/restructuring of helices (these may not be observed in MD simulations - time-scale etc). In the absence of a structure of CXE20 complexed with a ligand, some of the proposed model is conjecture. Something that can be addressed with discussion.

Minor comments

1) Introduction (lines 49-61), the authors could add a figure showing the chemical structure of SL to help the reader visualize the compounds used later.

2) Introduction (line 67), slow turnover rates ($\sim 0.33 \text{ min}^{-1}$) - instead of 1 molecule per 3 min. Better to provide kinetic context.

3) Introduction (lines 70-80), for completeness the authors include mention of other examples of phytohormone modification. There are multiple enzyme families that do this. For example, the SABATH methyltransferases (SA, IAA, and JA - Piotrowska & Bajguz, 2011 *Phytochemistry* 71, 2097; Qin et al 2005 *Plant Cell* 17, 2693) and the GH3 acyl acid amido synthetases that conjugate IAA with acidic amino acids to form inactive conjugates (Jez 2022 *Current Opin Plant Biology* 66, 102194).

4) Introduction (line 107) - delete “high” and replace with 1.8-2.3Å resolution - ‘high’ is relative.

5) Results (line 141) - the sequence analysis shows variability in the N-terminal regions of the CXE - are there organelle localization sequences in the N-terminal sequences? This could account for some of the variability. Please clarify.

6) In the kinetics tables of Figures 2 and 3 - swap the order of k_{cat} and K_m (should also have the k and K in italic) and provide errors for the values shown (not needed for k_{cat}/K_m). Similarly, in the text of the manuscript - fix K_m (italic K and subscript m).

7) Results (line 245 and 248) - 'strikingly' - used twice, perhaps vary word choice.

8) Results (line 289) - the comparison of catalytic efficiency needs to consider errors. Within error of experiments, the two values are comparable.

9) Table 1. The authors need to use significant figures. For cell dimensions, 83.92 83.82 117.2; can delete angles because these are defined by the space group. Resolution 48.0-2.30. R-factors - one decimal place is sufficient - same for B-factors. Are the unique reflections for data collection and refinement the same? Also, please include Ramachandran information.

Point-by-point response to reviewer's comments

We appreciate the reviewers for their efforts and constructive comments on our manuscript. Below is our point-by-point response to reviewers' comments.

Reviewer #1:

Phytohormones levels are regulated through specialized enzymes, participating not only in their biosynthesis but also in post-signaling processes for signal inactivation and cue depletion. CXE15 and CXE20 carboxylesterases have been shown to deplete strigolactones (SLs) that coordinate various growth and developmental processes and also function as signaling molecules in the rhizosphere. In this study, the authors elucidate the crystal structures of CXE15 (both apo and SL intermediate bound) and CXE20 that revealed new insights into the mechanisms of SL perception and catabolism. The N-terminal region of CXE15 and CXE20 has distinct secondary structures that play pivotal roles in their function both in vitro and in planta. Their findings indicate that a dynamic transition of the open-closed N-terminal helix domain in CXE15 facilitates robust SL hydrolysis. The result not only illuminates the distinctive process of phytohormone breakdown but also uncovers a novel molecular architecture and mode of plasticity within a specific class of carboxylesterases. The manuscript is well written and the conclusion is reliable. However, I suggest the following minor corrections should be made before the manuscript can be accepted for publish.

1. The authors failed to get the AtCXE15 crystal structure resolved by the full length of CEX15 alphafold model, however, after deletion of a truncated (1-50aa) alphafold search model, they were successful in getting the crystal structure of full length CEX15. In the crystal structure of CEX15, a cap domain (residues 1-21) and a core catalytic domain (residues 32-329) had covered the residues 1-50. Since there is a deletion of residues 1-50, how can the authors resolved the crystal structure without these residues?

[Response] We appreciate the reviewer for bringing up this point. It is indeed possible to determine X-ray crystallography structure solutions via Molecular Replacement methodology with as little as 30% structural identity of the search model (for further technical explanation, refer to doi: 10.1107/S2053273316010731, the Phenix-online.org documentation for MR solutions, and our explanation below). However, attempting to place a full-length search model that does not share the complete 3D fold identity with the structural solution may result in failure to achieve a reasonable solution. In our approach, we utilized a truncated model to solve the phase and subsequently rebuilt the full-length model with high accuracy based on the differential (Fo-Fc) electron density maps. The R-values obtained from our refinement process clearly indicate that our structure adheres to the best crystallography standards (refer to Table 1). We realized that our technical description in the main text of the previous version regarding the solution determination process was under the assumption that it may be common knowledge within the structural biologist community but perhaps not among general readers. To address this, we have removed this technical explanation from the results section and provided a clearer, elaborated explanation to describe in detail how we obtained the experimental/crystal structure for full-length CXE15. This can now be found in the *Methods* section under "*Crystallization, data collection, and structure determination.*" See lines 509-521.

It is worth noting that we find this point particularly intriguing in light of recent developments, especially with the emergence of AI/AlphaFold predictions, which rely on machine learning

algorithms trained on experimental structures. While we are keen to delve deeper into this phenomenon during our structure determination process, which uncovered a mis-predicted region by AI, we believe that it falls outside the scope of this current work, aside from providing a brief explanation of our solution process. However, we have included a more detailed explanation here for the reviewer, outlining the steps involved in the 3D structure solution process using search models to solve the phase, as well as our rationale for truncating and rebuilding the model.

We employed the standard Phenix-Phaser MR (molecular replacement) program, which is implemented in the Phenix software suite. MR operates by utilizing a known search model to probe its presence within the experimental electron density map. It systematically conducts rotations and translations of the search model within the unit cell of the crystal lattice, calculating the correlation between the model and the observed electron density at each orientation. This iterative process is repeated across a range of orientations and positions to identify the optimal fit of the search model within the crystal. Additionally, various scoring functions are computed to evaluate the quality of the molecular replacement solution at each iteration.

Once a potential solution is identified, Phaser initiates cycles of refinement to enhance the alignment between the model and the experimental data, generating differential electron density maps (FoFc and 2FoFc). Following the initial solution, the resulting model can undergo further refinement and rebuilding using tools such as Coot and Phenix.refine. This refinement process entails adjusting atomic coordinates, optimizing geometry, and iteratively fitting the model to the electron density map to improve the overall agreement between the model and the experimental data. A significant difference was found in the N-terminal region between the AlphaFold-predicted model of CXE15 and the experimental X-ray crystal structure data, as illustrated in the image below:

Images for the Reviewer

Therefore, in our subsequent search, we omitted the first 50 amino acids from the AlphaFold model (see the differences above). This truncated model yielded improved correlation and scoring functions, as evidenced by top LLG/TFZ solutions (showing above the 'missing' density as 2FoFc map in green). On the left below is the result obtained with AlphaFold, which achieved a solution but lacked accuracy upon inspection, while on the right is our truncated model, displaying high

scores and accuracy, albeit with only density for the missing N-terminal region (as shown above). The resulted structure solution involved manual building of the first 50 amino acids in Coot to achieve a fully complete model. Subsequently, the model was further refined, resulting in final excellent R_{work} and R_{free} values of 20% and 24%, respectively.

Image for the Reviewer: MR solution from phenix.phaser when full length AlphaFold model (left) and truncated AlphaFold model (right) was used

2. There were several methods to establish the model, such as selenomethionine-modified structure. Can author confirm the reliability of structure predicted by alphafold?

[Response] We have addressed this concern sufficiently above in response to reviewer's concern #1. Our structure solution was effectively achieved using the Molecular Replacement (MR) methodology, rendering experimental phasing unnecessary with heavy atoms/isotopes such as selenomethionine for MR-Single Anomalous Dispersion (SAD) or Multi-wavelength Anomalous Dispersion (MAD). MR of the truncated AlphaFold model led to a successful structure solution, clearly revealing electron density for all "missing" residues within the N-terminal region. The differential electron density accurately reflected the full-length CXE15, which was comprehensively built and refined, yielding outstanding R-values. The AlphaFold probability score was notably high (exceeding 90% probability based on similar existing structures). It is important to note that while our structure closely resembles the AlphaFold prediction altogether, the N-terminal region appears to be unique. Overall, the reliability of our structure, determined using experimental data, is of high accuracy and aligns with the best crystallography standards, as indicated by the values in Table 1 and the attached Validation Reports from RCSB-PDB. These reports undergo stringent validation by annotators who utilize all experimental density maps and beamline detectors values as part of the requisite verifications during the deposition process of our crystal structure data.

3. How about the similarity of AtCXE15 crystal structure compared to CEX crystal structure from other species?

[Response] This is an excellent suggestion from the reviewer, and we appreciate the opportunity to provide additional comparative insights between CXE15 and the carboxylesterase family within a broader context. To address this suggestion, we have added an additional figure (Supplementary Figure 6) where we analyze existing experimental crystal structures from plants (such as kiwi CXE1 and rice CXE10/CXE14 and CXE19/GID1a/b/c) and animals (including human CES1 and mouse CES2) in comparison with the CXE15/CXE20 fold. Our new comparative structural analysis reveals that while the N-terminal region of plant carboxylesterases exhibits significant variation, the core catalytic region remains highly conserved. Similarly, in animal carboxylesterases, the core catalytic region is conserved, but both mouse and human enzymes feature additional regulatory domains and a/b domains that are proposed to control the

entry of specific ligands. These results from our comparison are now also described in the Results section; see lines 181-193. Altogether, this comparative analysis underscores, yet again, the significant regulatory function of the N-terminal region in these enzymes. Our study clearly describes the first structural modulation by this class of CXE in plants.

4. *There is a lack of the phenotypes for the mutagenesis of residues that are critical for GR24 binding in AtCXE15.*

[Response] We thank the reviewer for highlighting this aspect. Indeed, the binding residues of SL/GR24 will be within the catalytic sites (e.g., Ser169 and Glu271). To address this, we have analyzed the phenotype of *CXE15*^{S169A/E271A} catalytic mutant through transient overexpression (OE) in *Nicotiana benthamiana* plants. The observed phenotype distinctly indicates a lack of axillary bud outgrowth, unlike in *CXE15*-OE, thereby confirming the essential role of the active protein in effectively catabolizing SLs in planta. In addition, our results show that expression of NbBRC1 is increased in *CXE15*^{S169A/E271A} catalytic mutant compared to *CXE15*. These new results have been included in Supplementary Figure 8b-c and in the results section of the manuscript; see lines 280-288.

5. *Are there any other methods to test the binding affinity between GR24 and AtCXE15/20 instead of DSF analysis and limited proteolytic digestion?*

[Response] Besides DSF and limited protein digestion, we conducted an analysis on the kinetics of *CXE15* and *CXE20* towards YLG, as indicated in Figure 2. YLG contains an active D-ring akin to that found in natural strigolactones and has been widely used to study SL/GR24 receptors/hydrolase catalytic rates (Tsuchiya et al., Science 2015, Shabek et al., Nature 2018). YLG incorporates a pro-fluorescent probe attached to its active D-ring. When acted upon by enzyme CXEs, these molecules undergo cleavage, releasing fluorescence that facilitates the measurement of enzyme kinetics. Because of the fairly rapid cleavage/processing of GR24 molecule it has been challenging to provide direct accurate kinetics measurement of natural strigolactones or synthetic hormones like *rac*-GR24. Therefore, the use of YLG enabled successful determination of biochemical function as well as enzyme kinetics, including affinity, K_m , and k_{cat} of the reaction, all of which are depicted in our

Figure 2 and Figure S9. In that regard, we have elucidated the crystal structure of *CXE15* in complex with the D-ring of *rac*-GR24, providing further insights into our findings. To further address this point, it should be noted that in our revised work we provided additional nano LC-MS/MS on the modified amino acids of *CXE15/20* (see **new Supplementary Fig. 10**). See the text in the results section, lines 335-346. This data supports the intermediate product of SL during the catalytic process by the CXEs.

6. *Supplementary Figure 3, the author should marked mAU with 280.*

[Response] We have edited the Y axis label to “UV Absorbance (280nm)” as suggested by the reviewer.

7. *line 474, The initial structure was immersed in a periodic water box of cubic shape (1.0 nm). This solvent box is too small, and the result is not reliable.*

[Response] We thank the reviewer for bringing to our attention this typo error. We have corrected this in the revised version and provided a clear explanation of the methodology used for running the dynamics simulation. Additionally, in the *editconf* command, we included the options "-c", "-d 1.2", and "-bt cubic", ensuring a minimum distance of 1.2 nm between the solute to the box edge, thus preventing the protein from interacting with its periodic image. This has been better described in the text under *Methods*; see lines 525-539.

8. *line 480, We subsequently applied LINCS64 constraints for all bonds, keeping the whole protein molecule fixed and allowing only the water molecule to move to equilibrate with respect to the protein structure. The procedures are not clear. Are there any restraints for the minimization and all equilibration steps?*

[Response] We thank the reviewer for pointing it out. We have now provided a clear explanation of the methodology utilized for running the molecular dynamics. During the equilibration phase of both the NVT and NPT ensembles, positional restraints were implemented to adjust the arrangement of solvent and solute molecules relative to the Protein or Protein-ligand complex. Subsequently, during the production phase of the simulation, these positional restraints were lifted. Apart from these positional restraints, no other constraints were employed throughout the simulation process. This has been better clarified in the text under *Methods*; see lines 525-539.

Reviewer #2:

This manuscript describes the strigolactone (SL) catabolism mechanism by carboxylesterases based on those crystal structures. CXE15 and CXE20 have been proposed to catalyze SL hydrolysis, which is possibly the active hormone degradation step. The authors of this manuscript successfully solved the crystal structures of CXE15 and CXE20 at high resolution and proposed that C-terminal helix region in CXE15 has a critical role in its catalytic function. The results of this manuscript are clear and reliable. However, I think the results of this manuscript do not have enough general interest to attract the readers of this journal. The biochemical function of CXE15 as the SL hydrolyzing enzyme has been proposed in this manuscript and a previous report, however, I think the scientific evidence is still not enough to prove the involvement of this enzyme in the SL deactivation process in vivo. The authors mention that ‘cxe15 mutants were more sensitive’ by citing a previous report (Enjun Xu et al, Nat Plants, 2012). However, this result was obtained by a hypocotyl elongation assay. If considering that the main physiological role of SL is shoot branching regulation, such data is not enough to conclude the involvement of CXE15 in SL catabolism in vivo. Moreover, there were no data directly showing that endogenous SL levels are significantly decreased in the CXE15 overexpressing plants. I do understand that the structural characterization of these enzymes is quite valuable in the field, however, it is likely to be of interest only to people in the limited field. Some major and minor comments are described below.

We thank the reviewer for taking the time to review our work and for providing valuable

suggestions for improvement. Our study builds on previous physiological evidence identifying CXE15 and CXE20 as key components in SL catabolism in planta. It should be noted that the previous study in *Nature Plants* (Xu, et al 2021) monitored not only hypocotyl elongation but also branching phenotypes and transcriptomic data of SL-depleted plants. We have clarified this in our responses below and provided additional data. Here, we elucidate and propose a molecular mechanism for SL phytohormone catabolism that has not been previously described. Importantly, our study, particularly in its revised form, offers a broader perspective on the carboxylesterase superfamily, which plays fundamental roles in numerous essential metabolic pathways across all kingdoms of life. We reveal a novel mode of action driven by these enzymes' molecular architecture and intrinsic dynamics, which has not been observed before. This adds an important layer of complexity to the understanding of these crucial enzymes, not only in plants but also in general carboxylesterases.

Nature Communications features numerous publications on similar themes, focusing on the structural and functional analysis of novel and unique modes of enzymatic regulation. Examples include Wang et al., 2023, on *catalytic site flexibility in Vibrio dual lipase/transferase*; Lopez-Alonso et al., 2022, on the *cryo-EM structural exploration of catalytically active pyruvate carboxylase*; and Perez-Garcia et al., 2021, on the *structure of a promiscuous ancestral enzyme in the metallo- β -lactamase family*.

Major comments

1. line; 74-75, CYP707A is not involved in the Jasmonoyl-L-isoleucine deactivation. The authors should correctly explain this part.

[Response] Thank you for bringing this to our attention. Indeed, this is an error and we have now changed the text to the correct CYP (members of the CYP94 cytochrome P450) involved in the deactivation of JA. See lines 60-61.

2. line 269-274. To conclude that the branching phenotype in the CXE15 expressing plant is due to SL degradation, the authors should analyze the endogenous levels of SLs.

[Response] Unfortunately, detections of endogenous levels of SLs in planta has been incredibly challenging in our field. To the best of our knowledge and personal communications, methods for detecting SLs in tissues are still under development and not yet available. SLs basal levels are already at fairly low concentrations in both plants and soil (fmol/g of fresh root weight and pmol/l of root exudate, see Yoneyama et al., Bio-Protocol, 2016), which added to the difficulties in detecting dynamic flux or catabolism events at the tissue levels. Also, compared to other phytohormones, SL detection is even more challenging due to its intrinsic fragility and non-canonical perception mechanism (slow hydrolysis by the D14 receptor-hydrolase). The intrinsic instability of SLs is attributed to the dilactone ring (see the image below: C- and D-ring of GR24 or any natural canonical SLs) connected by the enol-ether bridge, which dissociates in aqueous alkaline solutions. Despite the ability to detect SLs using high-performance liquid

chromatography-tandem mass spectrometry (HPLC-MS/MS), obtaining accurate results requires highly pure samples and substantial amounts of starting material, as a significant portion of the analyte material is lost during the extraction procedure. This adds further complexity to the detection and measurement of SL levels in planta (see also Floková et al., *Plant Methods*, 2020, and additional references below).

1. Floková, K., Shimels, M., Andreo Jimenez, B. et al. An improved strategy to analyse strigolactones in complex sample matrices using UHPLC–MS/MS. *Plant Methods* 16, 125 (2020). <https://doi.org/10.1186/s13007-020-00669-3>
2. Yoneyama K, Xie X, Nomura T, Yoneyama K. Extraction and measurement of strigolactones in sorghum roots. *Bio-Protocol*. 2016;2016:6.
3. Sato D, Awad AA, Chae SH, Yokota T, Sugimoto Y, Takeuchi Y, et al. Analysis of strigolactones, germination stimulants for *Striga* and *Orobancha*, by high-performance liquid chromatography/tandem mass spectrometry. *J Agric Food Chem*. 2003;51:1162–8.
4. Boutet-Mercey S, Perreau F, Roux A, Clavé G, Pillot JP, Schmitz-Afonso I, et al. Validated method for strigolactone quantification by ultra high-performance liquid chromatography—electrospray ionisation tandem mass spectrometry using novel deuterium labelled standards. *Phytochem Anal*. 2018;29:59–68.
5. Halouzka R, Tarkowski P, Zwanenburg B, Čavar Zeljković S. Stability of strigolactone analog GR24 toward nucleophiles. *Pest Manag Sci*. 2018;74:896–904.

Nevertheless, to address this important point, we have included additional explanations and evidence in the text and under *Introduction* section; see lines 75-86. We elaborated on the previously published work by Xu et al. (2021), which collected transcriptome data and measured upstream and downstream genes in the SL-mediated pathway, as well as monitored the branching phenotype of overexpressed *AtCXE15*. As shown on the right, *AtMAX3* and *AtBRC1* have been significantly upregulated and downregulated, respectively, in response to the overexpression of *CXE15* in *Arabidopsis*.

Moreover, we conducted additional experiments to investigate the catalytic function of the overexpressed *CXE15*^{S169A/E271A} catalytic mutant in *Nicotiana benthamiana* leaves in the revised version. The observed phenotype distinctly indicates a lack of axillary bud outgrowth, unlike in *CXE15-OE* plants, thereby confirming the essential role of the active protein in effectively catabolizing SLs in planta. These results have been included in the main text and added **new data in the Supplementary Figure 8**. Importantly, we also measured the *BRC1* transcript levels, which are known to be downregulated at depleted levels of SL as part of the downstream regulated SL-signaling genes. Indeed, in our *CXE15-OE* plants, but not in the *CXE15*^{S169A/E271A} catalytic mutant, *BRC1* was downregulated, further supporting the depletion of SL by *CXE* activity in planta and corroborating previous reports (e.g., Seale et al., *Development*, 2017; Xu et al., *Nature Plants*, 2021). See Supplementary Figure 8 and the results section, lines 280-288.

3. line 308-322, *The authors solved the crystal structure of the SL D-ring bound form of CXE15. Moreover, a unique reaction mechanism was proposed based on this intermediate structure. However, it is unlikely that the structural data is clear enough to get such a conclusion. To prove such a reaction mechanism proposed in Fig 4f, the authors should provide MS spec data which show that the entire GR24 molecule is*

covalently linked with CXE15. I do not understand how the water molecule is activated when it attacks the covalently linked intermediate.

[Response] We thank the reviewer for raising this question and inspire us to take yet additional approach to support our findings in our high-resolution crystal structures. To address this, we initiated a collaboration with the Dr. Boyer (at Universite Paris-Saclay, CEA, CNRS) who is the world expert in strigolactone field using nano-LC-MS/MS. We conducted nano-LC-MS/MS analysis of CXE15 and CXE20 in the presence of *rac*-GR24. As anticipated and observed in our previous work on KAI2 hydrolase (Guercio et al 2022), CXE15 displayed no significant changes in the presence of the ligand due to a rapid hydrolysis *rac*-GR24, hindering our ability to capture the adduct on the catalytic serine.

Intriguingly, CXE20-GR24 exhibited a mass increase of 96 Da, corresponding to the D-ring of GR24. Subsequent trypsin proteolysis revealed the adduct on the catalytic triad H302, akin to the binding pattern observed in strigolactone receptors such as D14s. This finding may also elucidate the slower catalytic activity of CXE20. These new results are now included **in the Supplementary Figure 10** and described in the Results section, see lines 335-346.

*4. As mentioned above, I am still skeptical that these enzymes contribute to the deactivation of SLs. If CXE15 is indeed involved in the SL catabolism, the *cxe15* mutant should accumulate high levels of SLs. Moreover, introduction of full-length CXE15, but not its C-terminal helix truncated form, should complement this phenotype. These data would be necessary to conclude the physiological function of CXE15.*

[Response] We thank the reviewer for pointing this out. While our data and the scope of our work provide clear evidence using multiple approaches on the catalytic activity and hydrolysis of SLs by CXEs, particularly CXE15 with high catalytic rates, the overall physiological functions of CXE15 (including transcriptome and phenotypic data) in deactivating SLs in planta have been thoroughly investigated in the previously published study by Xu et al. (2021).

Our study focuses on delving into the mechanistic aspects of SL catalysis using biochemical and structural investigations. We identified a unique mode of structural dynamics in the N-terminal helix (not the C-terminal), a function we have investigated both in vitro and through transient expression using truncated and catalytic mutants. These findings have been further revised and added as the **new Supplementary Figure 8** (as addressed in the comments above).

We believe the source of possible confusion was our lack of detailed description of the previous physiological observations of CXE15 and the *cxe15* mutant in plants. To address this, we have provided several examples below and have clarified and elaborated on these details in the *Introduction* section to give readers a better understanding of CXE15 as the primary SL catabolizing enzyme. See lines 74-86.

Image for the reviewer
Adapted from Xu et al., 2021 Nature Plants

As can be seen above, the *cx15* mutant does not show any phenotype, possibly due to strong feedback loop inhibition or because SL levels in plants are also controlled by yet uncharacterized enzymes. Furthermore, the hyperbranching phenotype was observed only when *AtCXE15* was overexpressed (*AtCXE15-OE*) and was not reversed by grafting *AtCXE15-OE* shoots onto wild-type rootstock. This suggests that overexpression of *AtCXE15* in shoots alone is sufficient to induce hyperbranching, illustrating CXE15's capacity to modulate SL levels at specific sites. Similarly, the excessive branching phenotype of SL biosynthetic mutants *atmax4-1* and *atmax1-1* was not fully complemented by grafting onto *AtCXE15-OE* rootstocks. These experiments underscore that overexpression of *AtCXE15* in plants leads to decreased levels of potential SLs and their precursors, such as CL, CLA, or methyl carlactone. This also rules out the possibility that the hyperbranching phenotype in *AtCXE15-OE* plants is due to disturbed SL biosynthesis. Moreover, strict feedback regulation of SL biosynthesis has been reported in planta, and the abundant accumulation of endogenous SLs does not manifest in an obvious shoot branching phenotype. This might explain why disruption of *AtCXE15* does not significantly affect shoot branching.

Minor comments

1. The authors are using the words such as 'ligand' and 'perceive'. As far as I know these words are appropriate for explaining the receptor proteins. I think it would be better to use 'substrate' and 'bind/accomodate' instead.

[Response] We have replaced the words "ligand" and "perceive" in the manuscript with 'substrate' and 'bind' accordingly.

2. line 157-161. The authors should explain the process of the CXE structural analysis more correctly.

[Response] To clarify, we have elaborated on the structural analysis and solution determination process in the *Results* section. Additionally, we have relocated some technical details to the *Methods* section, where we provide a detailed description of how we solved the full-length crystal structure of CXE15 and CXE20. See lines 509-521 and 525-539.

Reviewer #3:

The manuscript by Palayam et al presents a very complete structural and biochemical study on two carboxylesterases (CXE15 and CXE20) from the model plant Arabidopsis thaliana and their role in catabolism of the phytohormone strigolactone. This is a well performed, experimentally rigorous, and clearly presented study that provides new insights on strigolactone homeostasis.

We appreciate the reviewer's positive comments on our manuscript and on the importance of our work providing new insights on strigolactone homeostasis.

Major comments

1) In the title (and elsewhere in the manuscript), the authors use 'allosteric' regulation - this isn't accurate. Allostery implies an effector molecule that modulates (positive or negative) catalytic function at the active site. Perhaps 'conformational' regulation may more clearly fit what is observed in the x-ray crystal structures of CXE15 and CXE20.

[Response] We are grateful to the reviewer for highlighting this important notion. Consequently, we have **revised the title** of our manuscript to "**Conformational Regulation**" as a more fitting term to depict the observed modulation of CXEs.

2) At various points the abstract needs to provide more specific information for the reader. Line 25/26 - what organism? Perhaps add "Arabidopsis thaliana" before "CXE15 and CXE20 carboxylesterases ..." to let reader know what plant. Line 28 (and elsewhere in the text), "x-ray crystal structure" instead of just "crystal structure". The statement that the distinct secondary structures of the N-termini play pivotal roles for in vitro and in planta activity is a bit vague. What data supports this? Line 32 ... reword to "... transition of the N-terminal helix domain of CXE15 between open and closed forms facilitates ..." Line 34 - "novel" - overused word - replace or delete (same for other places in the manuscript text).

[Response] We thank the reviewer for providing critical and much-needed textual clarifications and modifications. With certain constraints related to word count limits, **we have rewritten** all the points raised in the abstract and other sections of the text.

3) Results, line 157 - the fact that an AlphaFold model did not succeed in molecular replacement is not evidence (nor implies) a novel architecture. There are many reasons why a molecular replacement can fail. Consider removing some of this text.

[Response] Indeed. We thank the reviewer for this comment. To address this, **we have removed this technical explanation of MR solution from the Results section**. This information has been edited to better describe the structure determination process and in the revised version included in the *Methods* section under "*Crystallization, data collection, and structure determination.*" See lines 509-521.

4) In general, the experimental data provided provides new mechanistic insights on the CXE15 and CXE20 enzymes function. One question for the reviewers is the difference in N-terminal structure between the two enzymes. The movement of the helical cap of CXE15 is clear - the value of having apo and complexed versions of the structure is clear. Something for the authors to consider, are there other structural changes of the N-terminal region of CXE20 between open and closed forms - there are multiple examples of ligand binding triggering ordering/restructuring of helices (these may not be observed in MD simulations - time-

scale etc). In the absence of a structure of CXE20 complexed with a ligand, some of the proposed model is conjecture. Something that can be addressed with discussion.

[Response] Indeed, this is an important notion to discuss. The structural changes predicted by MD simulation may not capture all possible conformations under physiological conditions and in the presence of the substrate. We have now highlighted in the Discussion that other conformations may be possible and could be triggered by the perception and catalysis process. See lines 376-379.

We have also added a new video derived from the MD trajectory data to better illustrate the structural dynamics of CXE15 and CXE20 in the presence of *rac*-GR24 (new **Supplementary Movie 1**). Overall, we clarify that in CXE20, the binding pocket loop is stabilized after GR24 binding, and unlike CXE15, the N-terminal region does not undergo an open-closed transition. However, the loop connecting the N-terminal cap to the core moves farther from the catalytic core, increasing the overall cavity of the pocket. Lastly, in our revised work, we have also conducted nano-LC-MS/MS combined with trypsin digestion (new **Supplementary Fig. 10 and new Supplementary Fig. 11**), revealing additional details on our proposed mechanism for both CXE15 and CXE20, where an intermediate catalytic adduct is found on H302 in CXE20. All these edits and clarifications have been introduced in the text. See lines 258-261, 335-346, and 541-559.

Minor comments

1) Introduction (lines 49-61), the authors could add a figure showing the chemical structure of SL to help the reader visualize the compounds used later.

[Response] We have added the chemical structure of SL and the analog GR24 used in our study as Supplementary Figure 1a. We have also added more relevant references in the text for readers who would like to have more information on SLs chemical classification in plants. See line 41.

2) Introduction (line 67), slow turnover rates ($\sim 0.33 \text{ min}^{-1}$) - instead of 1 molecule per 3 min. Better to provide kinetic context.

[Response] We have edited this part accordingly and included kinetic information in the *Introduction* section; see line 52.

*3) Introduction (lines 70-80), for completeness the authors include mention of other examples of phytohormone modification. There are multiple enzyme families that do this. For example, the SABATH methyltransferases (SA, IAA, and JA - Piotrowska & Bajguz, 2011 *Phytochemistry* 71, 2097; Qin et al 2005 *Plant Cell* 17, 2693) and the GH3 acyl acid amido synthetases that conjugate IAA with acidic amino acids to form inactive conjugates (Jez 2022 *Current Opin Plant Biology* 66, 102194).*

[Response] This is an important comment, we thank the reviewer for bringing this to our attention. We have now included these enzymes in the *Introduction* section along with their relevant references. See lines 60-65.

4) Introduction (line 107) - delete “high” and replace with 1.8-2.3Å resolution - ‘high’ is relative.

[Response] We removed the word “high” in the Introduction Section.

5) Results (line 141) - the sequence analysis shows variability in the N-terminal regions of the CXE - are there organelle localization sequences in the N-terminal sequences? This could account for some of the variability. Please clarify.

[Response] This is a great suggestion. We have analyzed the sequence of all 20 CXEs and found that only CXE4 contains a transit peptide at its N-terminus. It is interesting to note that none of the clade I-III family shows the presence of any transit peptide nor localization sequences. We have now added this notion into the Results section which further confirms that the diversity in N-terminal regions is not directly linked to subcellular localization. See lines 139-141.

6) In the kinetics tables of Figures 2 and 3 - swap the order of k_{cat} and K_m (should also have the k and K in italic) and provide errors for the values shown (not needed for k_{cat}/K_m). Similarly, in the text of the manuscript - fix K_m (italic K and subscript m).

[Response] We have edited and corrected those terms and values accordingly in figures 2 & 3 including adding error values.

7) Results (line 245 and 248) - ‘strikingly’ - used twice, perhaps vary word choice.

[Response] We have replaced it with ‘interestingly’ in the text and went through the manuscript to avoid other such redundancies.

8) Results (line 289) - the comparison of catalytic efficiency needs to consider errors. Within error of experiments, the two values are comparable.

[Response] Indeed, we have now included appropriate standard deviation and corresponding error values in the catalytic efficiency and other related comparisons.

9) Table 1. The authors need to use significant figures. For cell dimensions, 83.92 83.82 117.2; can delete angles because these are defined by the space group. Resolution 48.0-2.30. R-factors - one decimal place is sufficient - same for B-factors. Are the unique reflections for data collection and refinement the same? Also, please include Ramachandran information.

[Response] We thank the reviewer for the helpful comment. We have incorporated all the suggestions in revised Table 1: angles in the cell dimensions were removed and significant figures for the decimals have been implemented for cell dimensions, Resolution, R-factors, and B-factors. Additionally, we have included the Ramachandran Plot analysis. Indeed, in our statistical summary the unique reflections were the same for data collection and refinement.

REVIEWERS' COMMENTS

Reviewer #1 (Remarks to the Author):

I looked over revised manuscript and the new additions now have satisfactorily addressed my previous concerns.

Reviewer #2 (Remarks to the Author):

The authors addressed almost all the comments from three reviewers, and I do understand the significance of uncovering the molecular architecture within a specific class of carboxyesterases. However, there are still unresolved issues. As for analyzing the endogenous levels of SLs in plant tissues, the authors claim that 'methods for detecting SLs in tissues are still under development'. This is not true. In some previous reports (Mashiguchi et al, PNAS, 2022, doi: 10.1073/pnas.2111565119; Yoneyama et al, Plant Direct, doi: 10.1002/pld3.219.;

Abe et al, PNAS, 2014, doi: 10.1073/pnas.1410801111.; Seto et al, 2014, doi: 10.1073/pnas.1410801111.), the endogenous CL, CLA, MeCLA, OH-MeCLA were analyzed from the Arabidopsis plant tissues (shoot part). Therefore, the method itself is available. Even though the method itself exists, I know it is not easy. However, if the authors want to conclude that CXE15 is a deactivating enzyme for SLs, analysis of those endogenous levels is necessary. If the authors can't try this experiment for some reason, I don't force the authors to do it. However, in that case, I strongly suggest they explain it in the text. Without such data, there would be no direct evidence that this enzyme is a deactivating enzyme for SLs. The additional results regarding the expressional analysis are supportive, but not conclusive. Anyway, a correct explanation should be added.

The results obtained by overexpressing CXE15 in *Nicotiana benthamiana* are interesting. However, I can't understand what happened in planta. *Agrobacterium* was infiltrated into leaves according to the method. However, axillary buds outgrowth is promoted. This suggests that the transiently expressed enzyme is transported to the bud part. Alternatively, it suggests that SL may be biosynthesized in the leaf and then transported to the axillary bud. In either case, we need to be cautious about interpreting the results, as there have been no such reports so far. We cannot rule out the possibility that CXE15 acted on some other molecule to promote axillary bud growth. Again, though, measuring endogenous SL would provide a definitive answer. In any case, the authors should also mention the possibility that SL decomposition is not a direct factor.

In conclusion, I do understand the novelty and significance of the part concerning the structural analysis, but we suggest the authors give a sincere explanation about the lack of direct evidence for CXE15 as the SL deactivating enzyme.

Reviewer #3 (Remarks to the Author):

The authors have very nicely and thoroughly addressed points raised in the initial review.

Point-by-point response to reviewer's comments

We appreciate the reviewers for their efforts and constructive comments on our manuscript. Below is our point-by-point response to reviewers' comments.

Reviewer #1

I looked over revised manuscript and the new additions now have satisfactorily addressed my previous concerns.

[Response] We thank the reviewer for taking the time to go over our revised study and delighted to hear that there are no additional concerns.

Reviewer #2

The authors addressed almost all the comments from three reviewers, and I do understand the significance of uncovering the molecular architecture within a specific class of carboxyesterases. However, there are still unresolved issues. As for analyzing the endogenous levels of SLs in plant tissues, the authors claim that 'methods for detecting SLs in tissues are still under development'. This is not true. In some previous reports (Mashiguchi et al, PNAS, 2022, doi: 10.1073/pnas.2111565119; Yoneyama et al, Plant Direct, doi: 10.1002/pld3.219.; Abe et al, PNAS, 2014, doi: 10.1073/pnas.1410801111.; Seto et al, 2014, doi: 10.1073/pnas.1410801111.), the endogenous CL, CLA, MeCLA, OH-MeCLA were analyzed from the Arabidopsis plant tissues (shoot part). Therefore, the method itself is available. Even though the method itself exists, I know it is not easy. However, if the authors want to conclude that CXE15 is a deactivating enzyme for SLs, analysis of those endogenous levels is necessary. If the authors can't try this experiment for some reason, I don't force the authors to do it. However, in that case, I strongly suggest they explain it in the text. Without such data, there would be no direct evidence that this enzyme is a deactivating enzyme for SLs. The additional results regarding the expressional analysis are supportive, but not conclusive. Anyway, a correct explanation should be added. The results obtained by overexpressing CXE15 in Nicotiana benthamiana are interesting. However, I can't understand what happened in planta. Agrobacterium was infiltrated into leaves according to the method. However, axillary buds outgrowth is promoted. This suggests that the transiently expressed enzyme is transported to the bud part. Alternatively, it suggests that SL may be biosynthesized in the leaf and then transported to the axillary bud. In either case, we need to be cautious about interpreting the results, as there have been no such reports so far. We cannot rule out the possibility that CXE15 acted on some other molecule to promote axillary bud growth. Again, though, measuring endogenous SL would provide a definitive answer. In any case, the authors should also mention the possibility that SL decomposition is not a direct factor.

In conclusion, I do understand the novelty and significance of the part concerning the structural analysis, but we suggest the authors give a sincere explanation about the lack of direct evidence for CXE15 as the SL deactivating enzyme.

[Response] We thank the reviewer for the important and helpful insights. To address this notion, we took the Reviewer suggestion, and have now added the following clarification in the Discussion section: “ Remarkably, overexpression of CXE15 (compared to truncated or catalytically inactive CXE15) in Nicotiana benthamiana resulted in axillary bud outgrowth with a distinct expression pattern of BRC1, similar to other SL-deficient mutants. While this data strongly supports the biochemical function and structure observed here, it raises intriguing questions about the potential transport of the transiently expressed enzyme to the bud and/or

the transport of SL to the axillary bud. Additionally, we cannot rule out the possibility that CXE15 may act on another molecule and/or SL precursor to promote axillary bud growth. Therefore, future studies combining endogenous SL detection with CXEs overexpression or knockout lines will clarify how these enzymes precisely fine-tune SL levels in planta. “

Reviewer #3

The authors have very nicely and thoroughly addressed points raised in the initial review.

[Response] We are delighted to hear that all points were thoroughly addressed.